# MicroRNA-934 is a novel primate-specific small non-coding RNA with neurogenic function during early development

**Kanella Prodromidou[1][†]\*, Ioannis S Vlachos[2,3,4,5][†], Maria Gaitanou[1], Georgia Kouroupi[1], Artemis G Hatzigeorgiou[3], Rebecca Matsas[1]\***

[1]Laboratory of Cellular and Molecular Neurobiology-Stem Cells, Department of Neurobiology, Hellenic Pasteur Institute, Athens, Greece; [2]Department of Pathology, Beth Israel Deaconess Medical Center, Boston, United States; [3]DIANA-Lab, Hellenic Pasteur Institute, Athens, Greece; [4]Harvard Medical School, Boston, United States; [5]Broad Institute of MIT and Harvard, Cambridge, United States

**Abstract** Integrating differential RNA and miRNA expression during neuronal lineage induction of human embryonic stem cells we identified miR-934, a primate-specific miRNA that displays a stage-specific expression pattern during progenitor expansion and early neuron generation. We demonstrate the biological relevance of this finding by comparison with data from early to mid-gestation human cortical tissue. Further we find that miR-934 directly controls progenitor to neuroblast transition and impacts on neurite growth of newborn neurons. In agreement, miR-934 targets are involved in progenitor proliferation and neuronal differentiation whilst miR-934 inhibition results in profound global transcriptome changes associated with neurogenesis, axonogenesis, neuronal migration and neurotransmission. Interestingly, miR-934 inhibition affects the expression of genes associated with the subplate zone, a transient compartment most prominent in primates that emerges during early corticogenesis. Our data suggest that mir-934 is a novel regulator of early human neurogenesis with potential implications for a species-specific evolutionary role in brain function.

\*For correspondence:
kprodromidou@pasteur.gr (KP);
rmatsa@pasteur.gr (RM)

[†]These authors contributed equally to this work

**Competing interests:** The authors declare that no competing interests exist.

## Introduction

Formation of the neocortex in mammals is a critical developmental milestone that entails expansion of diverse progenitors and generation of a vast array of neuronal types destined to occupy a highly organized structure (*Bystron et al., 2008*). Human embryonic and induced pluripotent stem cells permitting experimental access to neural cells and differentiation stages that are otherwise difficult or impossible to reach in humans, are an essential means for understanding human biology in health and disease.

During development, the emergence in human and non-human primates of a larger cortex with greater complexity involves highly derived features including expanded progenitor zones and a relatively protracted time course of neurogenesis reflected in tightly controlled, proliferative and/or neurogenic potential of progenitor cells (*Sousa et al., 2017*; *Otani et al., 2016*). Additionally, an early event in primate forebrain development is the appearance of a transient zone, the subplate (SP), below the emerging cortical plate layers (*Bystron et al., 2008*; *Silbereis et al., 2016*). The SP is a highly dynamic zone of the developing cerebral cortex that expands dramatically in human and non-human primates to become the largest compartment of the fetal neocortical wall (*Duque et al., 2016*). It is composed of a heterogeneous cell population that includes migrating and post-migratory neurons, and many ingrowing axons forming transient axon terminals (*Silbereis et al., 2016*). Interestingly, it comprises some of the earliest-born cortical neurons while disturbances in the SP

have been associated with neurodevelopmental disorders (*Hoerder-Suabedissen et al., 2013*; *Kostović et al., 2015*).

The molecular identity and behavior of cortical precursors and the generation of appropriate cell fates during corticogenesis is achieved through precise regulation of spatio-temporal gene expression involving conserved and divergent developmental programs (*Sousa et al., 2017*). miRNAs represent a powerful component of this regulatory machinery by promoting cell fate transitions in the human and non-human primate cortex (*Nowakowski et al., 2018*). Mature miRNAs are a class of short non-coding RNAs of ~22 nucleotides in length that mediate post-transcriptional regulation of gene expression through direct degradation of their target mRNA and/or suppression of translation. miRNAs bind to their mRNA-targets in a sequence complementary fashion while recognition is primarily determined through a 'seed' region in the miRNA (*Bartel, 2009*). Following evolutionary adaptations, more than 100 primate-specific and 14 human-specific miRNAs have been identified in the developing brain (*Hu et al., 2012*; *Berezikov, 2011*). It has been shown that certain primate miRNAs uniquely distinguish different regions within the proliferative cortical germinal zones, while their target genes are principally involved in the regulation of cell-cycle and neurogenesis as well as in human neurodevelopmental disorders (*Arcila et al., 2014*). Therefore species-specific features, including miRNAs, can be critical towards the elucidation of processes as intricate as human neural development, not the least because these may also relate to neuropsychiatric and behavioral pathologies bearing a neurodevelopmental origin.

To identify miRNAs with a key role in early human neural development, we investigated miRNA expression during neural induction of human embryonic stem cells (hESCs) and induced pluripotent stem cells (iPSCs), a stage marking the entry of multipotent ectodermal cells into the neural lineage (*Muñoz-Sanjuán and Brivanlou, 2002*) and characterized by neural progenitor cell (NPC) expansion and early neuron generation. Systematic analysis of miRNA expression profiling revealed that a primate-specific miRNA, namely miR-934 (miRBase v21, *Kozomara and Griffiths-Jones, 2014*), displays a striking stage-specific expression during neural induction. The significance of this expression pattern was validated by comparison with data from early to mid-gestation human fetal cortical tissue. Here we present functional evidence complemented by molecular data, demonstrating that mir-934 impacts on the progenitor-to-neuroblast transition. Further, the miR-934-mediated transcriptional neurogenic program affects subsequent neuronal differentiation processes and modulates the expression of SP-enriched genes.

## Results

### Transcriptome profiling for coding and small non-coding RNAs during directed human neuronal differentiation in vitro

We performed a directed differentiation protocol of hESCs/iPSCs towards hESC/iPSC-derived neural progenitors (NPCs) (13 days in vitro; DIV) and hESC/iPSC-derived neurons (48 DIV) (*Figure 1a*). We used the hESC line HUES6 (15) and iPSC lines [C1-1 and C1-2 clones derived from skin fibroblasts of a male healthy donor and PD1-1 and PD1-2 clones derived from skin fibroblasts of a male patient with familial Parkinson's disease (PD)] generated and characterized as previously described (*Kouroupi et al., 2017*). For directed differentiation (*Figure 1a*), hESCs/iPSCs with characteristic expression of the pluripotency state transcription factor Nanog were allowed to form embryoid bodies (0–6 DIV) (*Chambers et al., 2009*). Neural induction towards anterior neuroectoderm was initiated in the presence of Noggin and TGFβ inhibitor, which favor the generation and expansion of Pax6+/Nestin+ cortical progenitor cells (NPCs) and the emergence of DCX+ neuroblasts/early neurons (NPC stage, 6–15 DIV; *Figure 1a*). Cells were then directed to differentiate further into neurons (15–48 DIV), which became prevalent in culture at 48 DIV (*Figure 1a*). RNA sequencing of long and small transcripts was performed at distinct stages of directed differentiation corresponding to hESCs/iPSCs, hESC/iPSC-derived NPCs (13 DIV) and hESC/iPSC-derived neurons (48 DIV). Global expression profiling of hESCs/iPSCs and their differentiated NPCs and neuronal derivatives confirmed a reset in both mRNA and miRNA expression following fibroblast reprogramming (*Figure 1b–f*). Distinct within-stage groupings were observed in the correlation map and the heatmap performed using the 300 most variable long RNA genes, respectively (*Figure 1b,e*). As shown in the heatmap in *Figure 1g*, the pluripotency markers NANOG and OCT4 (POU5F1) are primarily

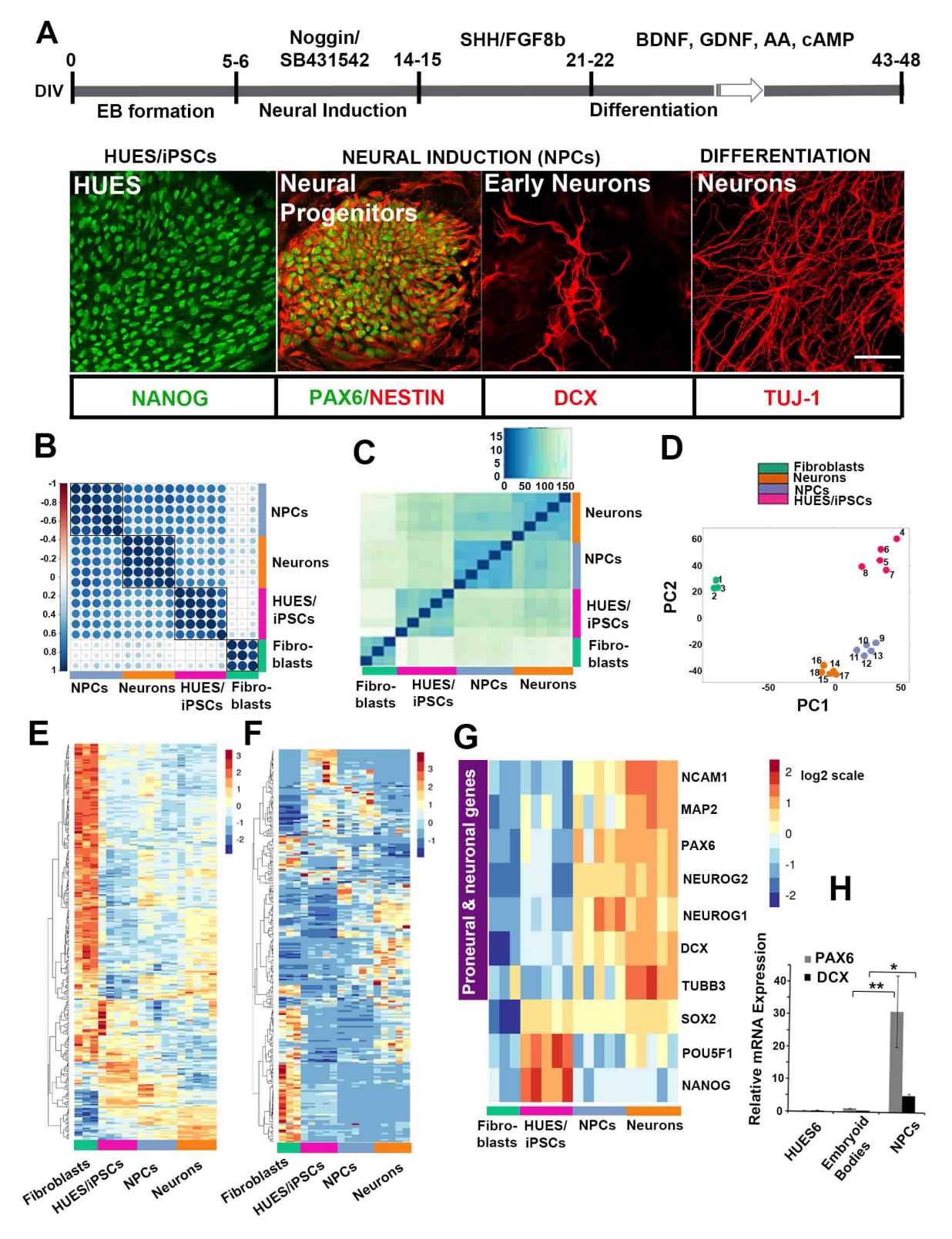

**Figure 1.** Directed neural differentiation of hESC/iPSC and RNA/small RNA sequencing at distinct stages of differentiation. (**A**) Schematic representation of directed neural differentiation of hESCs/iPSCs. Embryoid body (EB) formation (5–6 DIV) is followed by dual SMAD inhibition for neural induction and NPC generation (14–15 DIV). Neuronal differentiation then proceeds until 43–48 DIV in the presence of indicated factors. hESCs are positive for the pluripotency marker Nanog. At 12 DIV, characteristic rosettes comprising NPCs double-positive for Pax6 and Nestin are evident while

*Figure 1 continued on next page*

*Figure 1 continued*

DCX-positive neuroblasts/early born neurons are also seen. βIII-tubulin (TUJ-1)-positive neurons are predominantly detected at 48 DIV. Scale bar = 40 µm; (B) RNA expression correlation map incorporating the 300 RNA genes with highest standard deviations across all samples; groups exhibit distinct expression patterns and within-group uniformity. Darker blue colors depict stronger positive correlations between samples, as calculated by Pearson's correlation coefficient. (C) Heatmap depicting similarity between samples based on expression of the 50 most expressed miRNAs. Darker blue colors denote higher similarity. (D) Graph depicting the first two principal components (PC1, PC2) of the top 1000 miRNAs with the highest variance across all samples. All groups exhibit distinct expression patterns and within-group uniformity. Samples: Fibroblasts (green): 1, Fetal; 2. Control (C1); 3, PD-1; hESCs/iPSCs (magenta): 4, hESCs; 5, C1 iPSCs; 6, C2 iPSCs; 7, PD1 iPSCs; 8, PD2 iPSCs; iPSC-derived NPCs (purple): 9, hESC-NPCs; 10, C1 NPCs ; 11, C2 NPCs ; 12, PD1 NPCs; 13, PD2 NPCs; iPSC-derived neurons (orange): 14, hESC-neurons ; 15, C1 neurons ; 16, C2 neurons; 17, PD1 neurons; 18, PD2 neurons. (E) Heatmap depicting Z scores of log2-transformed expression levels of the 300 genes with highest variance across all samples. (F) Heatmap depicting Z scores of log2-transformed expression levels of the 250 miRNAs with highest variance across all samples. (G) Heatmap showing that expression of proneural and neuronal markers starts at the stage of neural induction. (H) RT-qPCR analysis shows up-regulation of Pax6 and DCX mRNA at NPC stage (n = 3, for Pax6 31.1±10.1 at the NPC stage; p=0.006 compared to HUES6, and for DCX 5±0.58 at the NPC stage; p=0.01 compared to HUES6). Bars and error bars represent mean values and the corresponding SEMs; 0.01<*p<0.05; 0.001<**p<0.01.

The online version of this article includes the following source data for figure 1:

**Source data 1.** qRT PCR data for PAX6 and DCX relative expression.

expressed in hESCs/iPSCs, while SOX2 has a more widespread profile, reflecting its association with both the pluripotency and NPC states. Expression of the neuronal precursor marker PAX6 and the proneural genes Neurogenins 1 and 2 (NEUROG1 and NEUROG2), characteristic of dorsal telenchephalon, and the early neuronal markers NCAM1 and DCX are already evident in NPC samples, while the neuronal marker MAP2 is most abundant at the more progressed neuronal differentiation stage. The up-regulated expression of PAX6 and DCX at the NPC stage was further validated by qRT-PCR (*Figure 1h*).

## Identification of miR-934 with species- and stage-specific expression during progenitor expansion and early neuron generation

Small non-coding RNA expression also characterized the stage of cell differentiation, as demonstrated by clustering of the 50 most expressed miRNAs, and the principal component analysis based on the expression of 1000 most variable miRNAs (*Figure 1c,d*). This result was confirmed by unsupervised machine learning (clustering of gene expression) for the 250 miRNAs with most variable expression (*Figure 1f*). Further examination of miRNA sequencing data verified that the miR-302–367 cluster is highly expressed in the pluripotency stage (*Figure 2a*; *Lipchina et al., 2012*), whilst miR-9 and miR-125 which are implicated in neuronal differentiation characterize primarily the progressed neuronal stage (the significance for differential expression was p<0.05 vs all other stages) (*Boissart et al., 2012*; *Delaloy et al., 2010*). On the other hand, miR-124 with a similar function in neurogenesis (*Yoo et al., 2009*) and abundant expression in the mature mammalian nervous system exhibited a more widespread expression pattern from NPCs to neurons (p<0.05 for neurons compared to fibroblasts and hESCs/iPSCs, and p=0.06 compared to NPCs) (*Figure 2a*). The mRNA and miRNA expression data derived from our analysis confirm the validity of our in vitro system to simulate distinct stages of human neurogenesis and differentiation.

To identify molecular mechanisms underlying early human neurogenesis, we looked for miRNAs associated with the stage of neural induction. To locate stage-specific miRNA regulators, we adopted the tissue-specificity index tau, which is a reliable method for estimation of gene expression specificity that has been shown to outperform other relevant indices (*Kryuchkova-Mostacci and Robinson-Rechavi, 2017*). Calculation of tau index across the distinct differentiation phases revealed that 144 miRNAs exhibited a highly specific expression profile (tau >0.7) (*Supplementary file 1*). For example, members of the miR-302–367 cluster, which is a known marker of pluripotency (*Lipchina et al., 2012*) was allocated at the hESC/iPSC stage (Fig, 2b and *Supplementary file 1*). When these 144 miRNAs exhibiting tissue specificity (tau >0.7) were ranked by decreasing order of median expression in NPCs, miR-934 emerged on top of the list, scoring the highest expression among other miRNAs at the NPC stage and a strikingly segregated expression as compared to all other stages (tau value 0.76) (*Figure 2b*; *Supplementary file 2*). Moreover, according to median expression differences, no other miRNA was identified to have such a clear disparity against other stages.

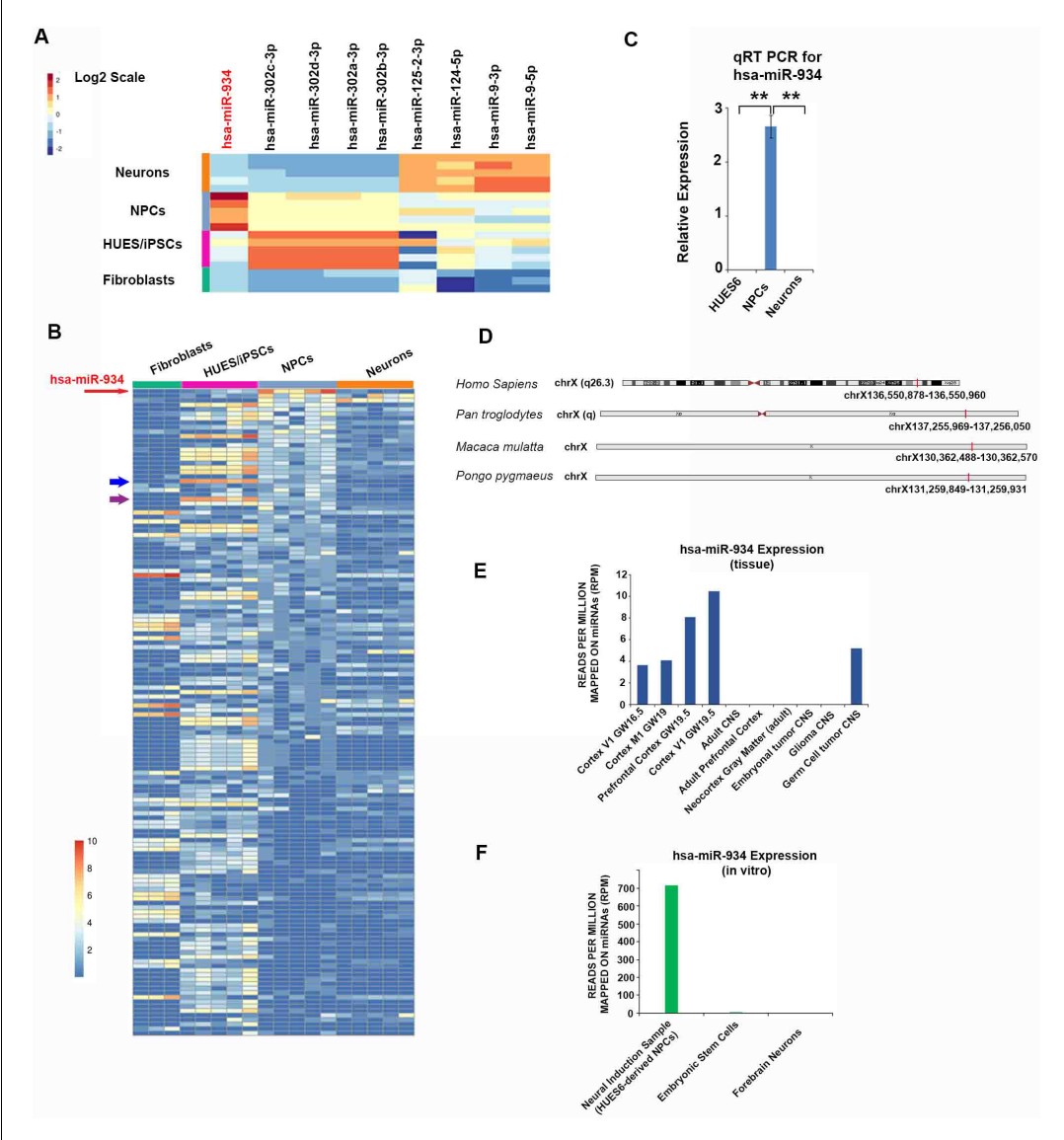

**Figure 2.** miR-934 is specifically expressed during progenitor expansion and early neuron generation. (**A**) Heatmap profiling the expression of selected characteristic miRNAs across directed neural differentiation of hESCs/iPSCs to NPCs and neurons, demonstrating also the high expression of miR-934 during neural induction (Fibroblasts vs NPCs; p=0.0004, iPSCs vs NPCs; p=0.0001, Neurons vs NPCs; p=3.5×10$^{-6}$). Warmer colors signify higher expression Z scores of the log2-transformed expression values. (**B**) Heatmap profiling the expression (log2; reads per million) of 144 miRNAs with stage specificity (tau >0.7) which are sorted based on median expression in NPCs. miR-934 is the highest expressed miRNA in NPCs and clearly segregated, exhibiting a stage-specific expression pattern. Red arrow denotes miR-934 ranking on the top, while warmer and cooler colors signify higher and lower expression values, respectively. The blue and purple arrows indicate hsa-miR-302d-5p and hsa-miR-302b-5p, respectively, which are selected in the HUES/iPSCs stage. (**C**) RT-qPCR analysis of hESCs and their NPC- and neuronal derivatives confirms the highly segregated expression of miR-934 at the neural induction stage (2.6±0.2, n = 3, p=0.006). (**D**) Chromosomal location of miR-934 in human and non-human primate species (adapted from UCSC Genome Browser. (**E, F**) Graphs incorporating miRNA expression data from uniformly-analyzed small RNA-Seq libraries (detailed in *Supplementary file 4*) corresponding to human fetal and adult neural tissue (**E**) and cellular samples with reference to our NPCs data (**F**). miR-934 expression shows selective expression in samples associated with human cortical development in vivo and neural induction in vitro. Bars and error bars represent mean values and the corresponding SEMs; 0.001<**p<0.01.

The online version of this article includes the following source data for figure 2:

**Source data 1.** qRT PCR data for hsa-miR-934 relative expression.

The high and segregated expression of miR-934 during neural induction was verified by RT-qPCR in hESC-derived NPCs. Concurrently, miR-934 expression was undetectable in hESCs and hESC-derived neurons (*Figure 2c*). In silico search and further analysis revealed that miR-934 is only annotated in certain primates, including humans (miRBase v21, June 2017); it has been experimentally identified in *Homo sapiens* and *Macaca mulatta*, while it has been predicted in *Pongo pygmaeus*, *Callithrix jacchus* (common marmoset) and *Pan troglodytes* (common chimpanzee) (*Figure 2d* for genomic location on chromosome X) (Appendix 1 for methodology of in silico analysis towards verification of miR-934 species conservation and *Supplementary file 3* for the respective results).

In order to define the expression pattern of miR-934, we analyzed a total of 58 small RNA-Seq libraries from various human tissues and cell samples (detailed in *Supplementary file 4*). The analysis confirmed a distinct expression signature for miR-934 which associates predominantly with neural tissues, particularly fetal cortical tissues (*Figure 2e* and *Appendix 1—figure 1* for non-neural tissue assessment). Specifically, a recent study reported consistent expression of miR-934, even if at low levels, during cortical development at 16.5–19.5 gestational week (GW) (*Nowakowski et al., 2018*; *Figure 2e*). According to meta-analysis of available data, we attest that miR-934 expression is undetectable in the adult nervous system (*Figure 2e*). Moreover, in agreement with our results, there is no expression of miR-934 in publicly accessible miRNA data from samples of embryonic stem cells and differentiated neurons (*Supplementary file 4* and *Figure 2f*). These findings further corroborated the tissue- and stage-specific expression of miR-934 and prompted us to investigate further its function.

## miR-934 controls the balance between Pax6+ progenitors and DCX+ neuroblasts and affects the morphological transition of newly born neurons

To gain insight into the physiological role of miR-934 in early human neural development, we examined the functional consequences of miR-934 modulation at the stage of neural induction of hESCs in vitro, using synthetic miRNA mimics or inhibitors to enhance mature miR-934 expression or block its function, respectively. Gain-of-function of miR-934 created a shift towards neurogenesis with the generation of an increased number of DCX+ neuroblasts/early neurons by 41.8% and a simultaneous reduction of PAX6+ progenitors by 39.4% (*Figure 3a–d* and quantification in *Figure 3o,p*). Conversely, inhibition of miR-934 action enhanced the number of PAX6+ precursors by 39.5%, while decreasing the levels of DCX+ neuroblasts/early neurons by 27.8% (*Figure 3e–h* and quantification in *Figure 3q,r*). This data suggests that miR-934 acts at the progenitor to neuroblast switch to promote differentiation.

As neuronal birth and differentiation are tightly linked processes, we next assessed the effect of miR-934 on the morphology of newly born neurons. We evaluated the length of the longest βIII-tubulin+ process as a measure of neurite outgrowth and the acquisition of a bipolar shape as an indication of neuronal morphological transition (*Pinheiro et al., 2011*). Measurements on DCX-positive neurons co-expressing the neuronal marker βIII-tubulin (TUJ1) revealed a 50% reduction in the length of the longest TUJ1+ neurite [*Figure 3i–n,s*; p=0.02], and a 40% decrease in TUJ1+ bipolar cells (*Figure 3i–n,t*; p=0.003) due to miR-934 inhibition. These results indicate that miR-934 controls neuronal birth and influences tightly linked differentiation processes in newborn neurons.

## The mRNA targets of miR-934 during neural induction are associated with progenitor proliferation and neuronal differentiation

To locate mRNA targets of miR-934 we explored the RNA-seq data obtained upon transition from hESCs/iPSCs to NPCs, as well as a second set of RNA-seq data generated at the stage of neural induction of hESCs following sustained inhibition of miR-934 function via transduction with the miR-Zip lentivector-based anti-microRNA system. To identify miR-934 targets in each case, we integrated small RNA and RNA sequencing data using the algorithm presented in mirExTra v2 (*Vlachos et al., 2016*). Targets were detected using microT-CDS target prediction tool as the source of potential interactions (*Paraskevopoulou et al., 2013*; *Supplementary file 5*).

The identified miR-934 targets comprise Frizzled 5 (FZD5) (microT-CDS prediction score 0.85), a member of the 'frizzled' gene family of receptors for Wnt signaling proteins, TFCP2L1 (microT-CDS prediction score 0.796), a stem cell self-renewal factor (*Ye et al., 2013*), as well as STMN2 (stathmin-

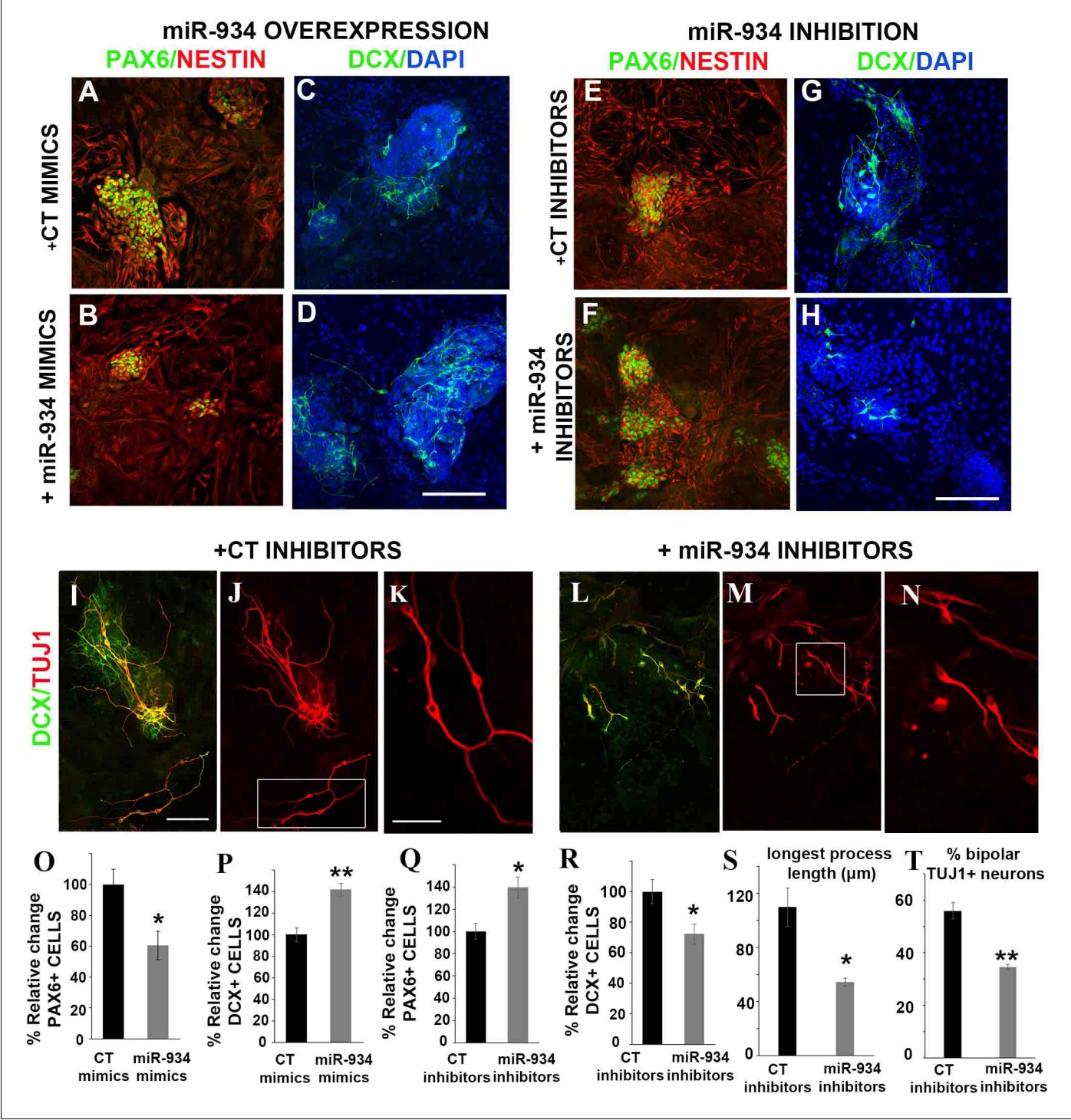

**Figure 3.** Perturbation of miR-934 affects the balance between Pax6+ progenitors and DCX+ neuroblasts and impacts on the morphological transition of newly born neurons. (A–H) Representative confocal images of hESC-derived cultures at neural induction stage (DIV12) following immunolabeling for Pax6 (green) and Nestin (red) or for doublecortin (DCX, green), upon miR-934 overexpression or inhibition, as indicated. Cell nuclei were visualized with Hoechst (blue). Scale bar = 100 μm. (O, P) Overexpression of miR-934 resulted in a reduction of the percentage of PAX6+ progenitors out of all cells in culture (CT mimics: 100 ± 9.9 and miR934 mimics: 60.6 ± 9.2, p=0.04, n = 3) and an increase of the percentage DCX+ neuroblasts out of all cells in culture (CT mimics: 100 ± 6.1 and miR934 mimics: 141.8 ± 5.6, p=0.007, n = 3). (Q, R) Inhibition of miR-934 has the opposite effect (for PAX6+% cell quantification: CT inhibitors 100 ± 7.2 and miR934 inhibitors 139 ± 9.6, p=0.01, n = 5; for DCX+% cell quantification: CT mimics 100 ± 7.6 and miR934 inhibitors 72.2 ± 6.5, p=0.02, n = 5–6). (I–J, L–M) Representative confocal images following immunolabeling for DCX (green) and TUJ1 (red) of control cultures at neural induction (DIV12) or upon miR-934 inhibition. Scale bar = 100 μm. TUJ1+ cells in white insets in (J, M) are shown at higher

*Figure 3 continued on next page*

Figure 3 continued

magnification in (**K, N**), respectively. Scale bar = 50 µm. Measurements performed were based on TUJ1 expression: Inhibition of miR-934 during neural induction affects the morphology of newly-born neurons leading to (**S**) reduction of the length of the longest TUJ1+ neurite (µm): for control inhibitors (CT) 109.78 ± 14. 2 vs. 54.53 ± 2.8 for miR-934 inhibitors; p=0.02, n = 4, 280 cells and (**T**) to decreased number of TUJ1+ neurons exhibiting a bipolar phenotype (% bipolar neurons over total neurons analyzed for control CT inhibitors: 55.9 ± 3.1 vs. 34.4 ± 1.1 for miR-934 inhibitors; p=0.003, n = 4, 160 cells). Bars and error bars represent mean values and the corresponding SEMs; 0.01<*p<0.05; 0.001<**p<0.01.
The online version of this article includes the following source data for figure 3:

**Source data 1.** Data on estimation of % Relative PAX+ and DCX+ cells.
**Source data 2.** Data on measurements of Longest Process in TUJ1+ neurons and of % of bipolar TUJ1+ neurons.

---

2, SCG10) (microT-CDS prediction score 0.853) and RAB3B (microT-CDS prediction score 0.755), which are implicated in neurite outgrowth and neurotransmission (*Riederer et al., 1997*; *Schlüter et al., 2004*; *Supplementary file 5*). We confirmed by qRT-PCR that sustained inhibition of miR-934 during neural induction up-regulates the four predicted targets FZD5, TFCP2L1, STMN2, and RAB3B (*Figure 4a–b*). The in silico analysis also predicted F11R a junctional adhesion molecule (*Ostermann et al., 2002*; *Sobocka et al., 2000*; *Fededa et al., 2016*) and SLC16A1 a monocarboxylate transporter (*Gerhart et al., 1997*; *Lauritzen et al., 2015*) as potential binding partners of miR-934, yet we could not confirm changes in their mRNA expression upon miRNA-934 perturbation (*Supplementary file 5*; *Figure 4a*–b).

The putative miRNA recognition elements (MREs) on the 3'-UTRs of the predicted targets FZD5, TFCP2L1, STMN2, and RAB3B contain binding sites with high probability for functional impact (8mer, 7mer and 9mer binding sites) (*Agarwal et al., 2015*). From an evolutionary point, it is worth noting that the identified 8mer binding site for STMN2 is expressed in various species whilst the respective MREs for TFCP2L1 and RAB3B are mainly conserved in primates (*Supplementary file 6*). To validate the bioinformatics data for target prediction we ran luciferase assays to selectively assess the interaction between miR934 and TFCP2L1 or RAB3B. To this end, we cloned regions of their 3'UTRs containing the individual binding sites for each target into the pmiRGLO luciferase reporter vector. Co-transfection of HEK293T cells with each of the modified vectors and miR934 mimics followed by measurement of luciferase activity confirmed the binding of miR934 at specific sites of these mRNAs (*Appendix 1—figure 2*).

Regarding the two predicted binding regions for miR934 on the 3'UTR for FZD5, the 7mer seed-matched site presents expression in different species while the 8mer seed-matched site is only conserved between human and the common chimpanzee (*Supplementary file 6*). Given the recognized role of Wnt signaling pathway during development and particularly in human progenitor cell proliferation and fate specification (*Woodhead et al., 2006*; *Yao et al., 2017*), we investigated if interaction of miR-934 with FZD5 can differentiate between the species-specific 8-mer and the more widely occurring 7-mer binding site. To this end, we examined independently the binding of miR-934 on each of the two predicted MREs using the pmiRGLO luciferase reporter system and determined that miR-934 indeed targets the 8mer seed-matched site, which is only conserved between human and the common chimpanzee (Appendix 1 for method and *Appendix 1—figure 3* for results). To validate this interaction, we cloned the respective 29nt-long MRE or its mutated form into the dual luciferase reporter vector (*Figure 4c–d*). Co-transfection of miR-934 mimics and the 8mer seed reporter construct suppressed luciferase activity in HEK293T cells by 36% (p=0.002), thus validating and specifying the region of interaction between miR-934 and FZD5 (*Figure 4d*).

Finally, we examined the effect of miR-934 on the Wnt pathway using immunoblot analysis (*Logan and Nusse, 2004*). Sustained inhibition of miR-934 during neural induction resulted in a moderate up-regulation of FZD5 protein and active β-catenin by 29 ± 4.5% and 28.5 ± 5.2% respectively (*Figure 4a,e–g*), suggesting that miR-934 acts to suppress β-catenin expression, a downstream effector of the Wnt pathway, by silencing FZD5 expression.

Taken together our data support that miR-934 acts during neural induction to suppress genes implicated in stem cell self-renewal and neurogenesis.

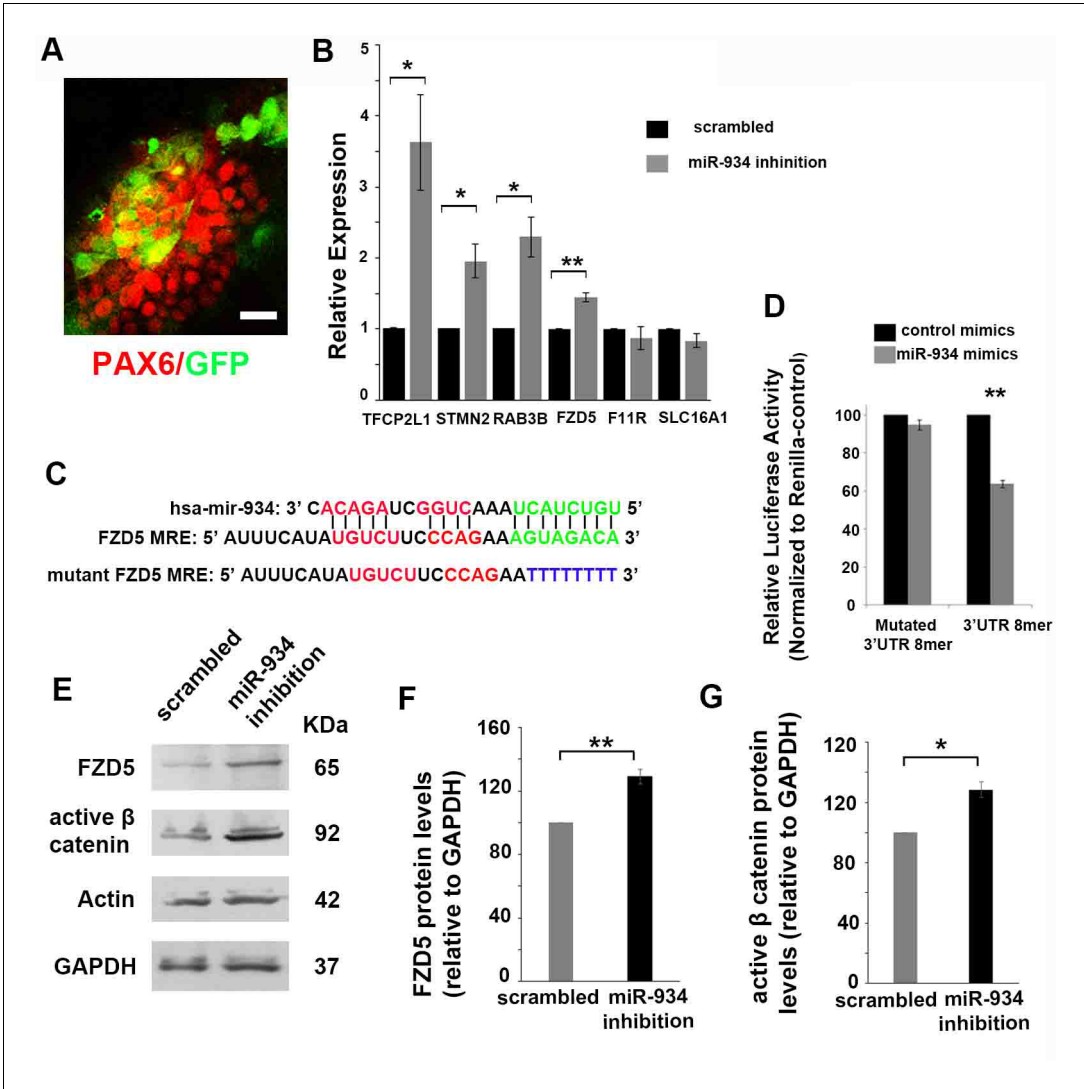

**Figure 4.** Identification of miR-934 targets during neural induction. (**A**) Confocal image showing PAX6+ neural progenitors (red) at DIV 15 transduced with miRZip lentivector (GFP fluorescence, green) for sustained inhibition of miR-934. Scale bar: 20 μm. (**B**) qRT-PCR data showing upregulation of miR-934 targets upon sustained miR-934 inhibition: TFCP2L1 (scrambled control 1±0.006 vs miR934 inhibition 3.63±0.67, p=0.03, n = 4), STMN2 (scrambled control 1±0.003 vs miR934 inhibition 1.95±0.23, p=0.02, n = 4), RAB3B (scrambled control 1±0.003 vs miR934 inhibition 2.29±0.28, p=0.01, n = 4), and FZD5 (scrambled control 1,004±0005 vs miR934 inhibition 1,44±0,061, p=0,005, n = 4). No changes were observed in the mRNA expression of F11R or SLC16A1 (p=0,18 for SLC16A1and p=0,47 for F11R, n = 4). (**C**) Sequence complementarity of miR-934 with the 8-mer binding site in Fzd5 3'-UTR region (green) and corresponding mutated site (blue), used in luciferase activity assay. (**D**) Estimation of the suppression of luciferase activity in HEK293T cells upon co-transfection with miR-934 mimics and the MRE (8-mer binding site) reporter construct. Suppression of luciferase activity was released upon mutation of the FZD5 8mer binding site (p=0.002, n = 3). (**E–G**) Sustained inhibition of miR-934 results in increased protein levels of FZD5 (p=0.007, n = 4) (**E, F**) and active β-catenin (p=0.01, n = 4) (**E, G**). Bars and error bars represent mean values and the corresponding SEMs; 0.01<*p<0.05; 0.001<**p<0.01.

The online version of this article includes the following source data for figure 4:

**Source data 1.** qRT PCR data for expression of miR-934 predicted targets upon sustained miR934 inhibition.
**Source data 2.** Data on luciferase activity in HEK293T cells upon co-transfection with miR-934 mimics and the MRE (8-mer binding site) reporter construct.
**Source data 3.** Data on western blot results for FZD5 and b-catenin proteins following miR934 Sustained inhibition.

# Inhibition of miR-934 during neural induction affects molecular pathways of neurogenesis and alters the expression of subplate-enriched genes

To further assess downstream signaling networks affected by miR-934, we examined the RNA profiling of hESC-derived cells at the neural induction phase upon sustained miR-934 inhibition. Differential gene expression analysis provided molecular data corroborating and extending our functional data. Overall miR-934 inhibition resulted in altered expression of 1458 genes (650 upregulated and 810 downregulated; p value<0.05), which according to GO (Gene Ontology) enrichment analysis (q value < 0.05) are primarily allocated to biological processes affecting progenitor cell proliferation, differentiation, axonal growth/guidance, neuronal migration, and neurotransmitter transport/release (*Figure 5a*). Moreover, the reported miR-934 targets STMN2 and RAB3B, match with the processes revealed by enrichment analysis, conforming to the categories of axonogenesis and neurotransmission (*Figure 5c,e*).

In terms of precursor-to-neuron differentiation, miR-934 inhibition coincides with up-regulation of N-Cadherin, which promotes self-renewal of NPCs (*Nowakowski et al., 2018*), induction of ID2 (inhibitor of DNA binding protein 2) that inhibits differentiation and promotes proliferation (*Paolella et al., 2011*), and reduction of GATA2, a transcription factor essential for neurogenesis (*El Wakil et al., 2006*; *Figure 5b,g*). With respect to axonal development, we detected miR-934-driven dysregulation of SYNGAP1 expression, an essential repressive factor involved in neuronal differentiation by controlling the timing of dendritic spine synapse maturation (*Kim et al., 2003*; *Rumbaugh et al., 2006*; *Vazquez et al., 2004*; *Figure 5c,g*). We also observed dysregulation of genes associated with tract formation and pathfinding, including molecules that belong to the semaphorin – neuropilin – plexin pathway, of their downstream intracellular mediators CRMPs (collapsing response mediator proteins), and also Ephrins and ROBOs (*Figure 5c*; *Polleux, 1998*). Moreover, there were deviations in expression of the Neuregulin-1/ErbB4 signalling partners, as well as of Reelin, the stimulus-regulated transcription factor Serum Response Factor (SRF) and the extracellular matrix protein matrilin 2, which are involved in neuronal migration (*Scandaglia et al., 2015*; *Figure 5d*). Proteins linked to neurotransmitter release that were affected include Snapin and members of Syntaptotagmin family (*Figure 5e*). Verification by qRT-PCR of the differential expression of selected genes is shown in *Figure 5g*.

Regarding cellular identities, analysis of differential data revealed altered expression of genes including TCFAP2C, MOXD1 and LEF1, a Wnt downstream target (*Behrens et al., 1996*) that characterize dorsal telencephalic radial glia and apical progenitors, as recently reported in single cell RNA-seq profiling of human fetal brain (*Nowakowski et al., 2018*; *Gan et al., 2014*; *Nowakowski et al., 2017*; *Figure 5f* and validation by qRT-PCR in *Figure 5g*; *Supplementary file 8*). On the other hand, we did not observe differences in the expression of the transcription factor Tbr2 (EOMES) that characterizes intermediate progenitors and which was found in particularly low abundance, or the outer radial glia marker HOPX (*Pollen et al., 2015*; *Appendix 1—figure 4a*). Moreover, there were no changes in ventral or mid-brain fate specification markers, including DLX1, DLX2, ALSC1, PAX2 and GBX2 which all exhibited lower expression as compared to the dorsal forebrain marker EMX2 (*Anderson et al., 1997*; *Casarosa et al., 1999*; *Harada et al., 2016*; *Desmaris et al., 2018*; *Appendix 1—figure 4b*). Of interest, we did not detect expression of early markers of lateral and medial ganglionic eminences, such as Nkx2.1, GSX1 and GSX2 with or without miR-934 perturbation (*Butt et al., 2008*; *Pei et al., 2011*).

To determine the developmental stage involved, we compared differential RNA expression data for genes that characterize the molecular profile of prodromal/pioneer neuronal populations during cortical development. We detected low expression in a series of early neuronal markers, while further analysis did not reveal miR-934-mediated differences in the expression of genes characterizing deep layer neurons, such as FEZF2, TBR1, FOXP2 and CTIP2 (BCL11B), or the early fate determinant of upper layer neurons CUX2 (*Molyneaux et al., 2007*; *Appendix 1—figure 4a*). We detected expression of SATB2 which is reported to be co-expressed with CTIP2 in a subset of early neurons of the visual cortex (*Nowakowski et al., 2017*; *Zhong et al., 2018*), yet this transcription factor presented no change upon miR-934 perturbation (*Appendix 1—figure 4a*).

We next examined subplate-associated genes as the SP is a major site of neocortical neurogenesis containing some of the earliest born neurons. For this comparison we assessed a series of

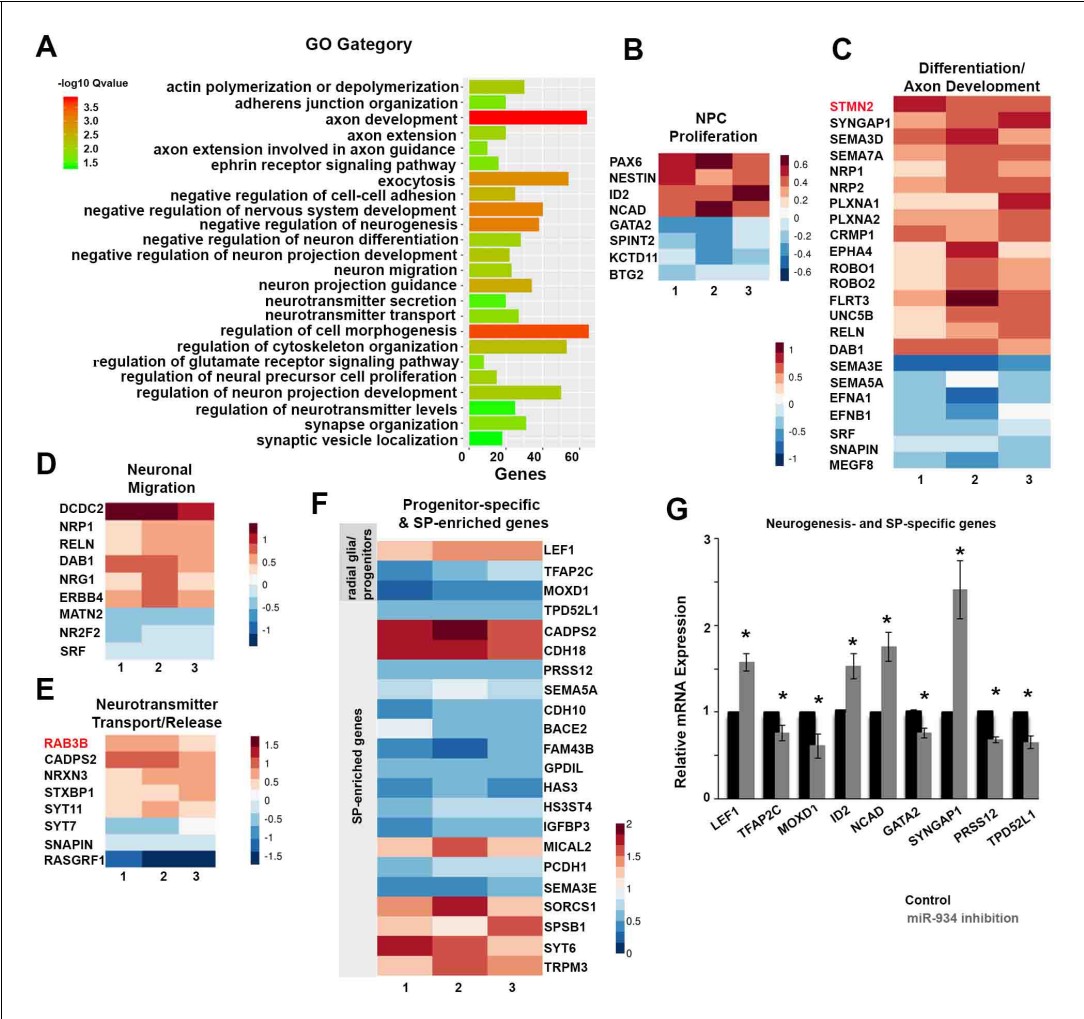

**Figure 5.** Sustained Inhibition of miR-934 induces global expression changes in genes associated with NPC proliferation and neuronal differentiation. (A) GO enrichment analysis of differential gene expression data following sustained inhibition of miR-934. (B–E) Heatmaps presenting the log2-fold change of molecules involved in the indicated biological processes following miR-934 inhibition, as determined by GO enrichment analysis, from three independent experiments 1, 2 and 3. miR-934 targets STMN2 and RAB3B are depicted in red. (F) Heatmap presenting the log2-fold change of progenitor-specific and SP-enriched genes. Warmer colors indicate higher fold changes. (G) Graph showing qRT-PCR validation data for genes affected by miR-934 inhibition (n = 3–4): LEF1 (scrambled control 1±0.0003 vs miR934 inhibition 1.57±0.09, p=0.009), TFAP2C (scrambled control 1±0.0006 vs miR934 inhibition 0.75±0.08, p=0.04), MOXD1 (scrambled control 1±0.0002 vs miR934 inhibition 0.6±0.14, p=0.04), ID2 (scrambled control 1±0.01 vs miR934 inhibition 1.52±0.14, p=0.03), NCAD (scrambled control 1±0.002 vs miR934 inhibition 1.75±0.17, p=0.01), GATA2 (scrambled control 1±0.01 vs miR934 inhibition 0.75±0.06, p=0.01), SYNGAP1 (scrambled control 1±0.0004 vs miR934 inhibition 2.4±0.33, p=0.02), PRSS12 (scrambled control 1±0.0002 vs miR934 inhibition 0.67±0.03, p=0.002) and TPD52L1 (scrambled control 1±0.0002 vs miR934 inhibition 0.65±0.07, p=0.03). Bars and error bars represent mean values and the corresponding SEMs; 0.01<*p<0.05.

The online version of this article includes the following source data for figure 5:

**Source data 1.** qRT PCR data for expression of genes affected by miR-934 inhibition.

subplate-enriched molecular markers with reference to the RNA-seq dataset generated by Miller et al from human fetal brain at gestational weeks 15 and 16 (*Miller et al., 2014*; *Supplementary file 8*; *Figure 5f*). We observed alterations in 19 subplate-enriched genes, including two genes PRSS12 and TPD52L1 identified by *Hoerder-Suabedissen et al., 2013*; *Luhmann et al., 2018* that were validated by qRT-PCR (*Figure 5g*).

Finally, miRNA profiling showed significant changes in 79 miRNAs (p value<0.05) in response to miR-934 inhibition. Bibliographic examination of the 20 differentially expressed miRNAs exhibiting the lowest p values revealed their association with axon/neurite growth and synapse formation (*Figure 6a,b*). Further we identified miR-34c-5p, which plays an important role in neurogenesis and

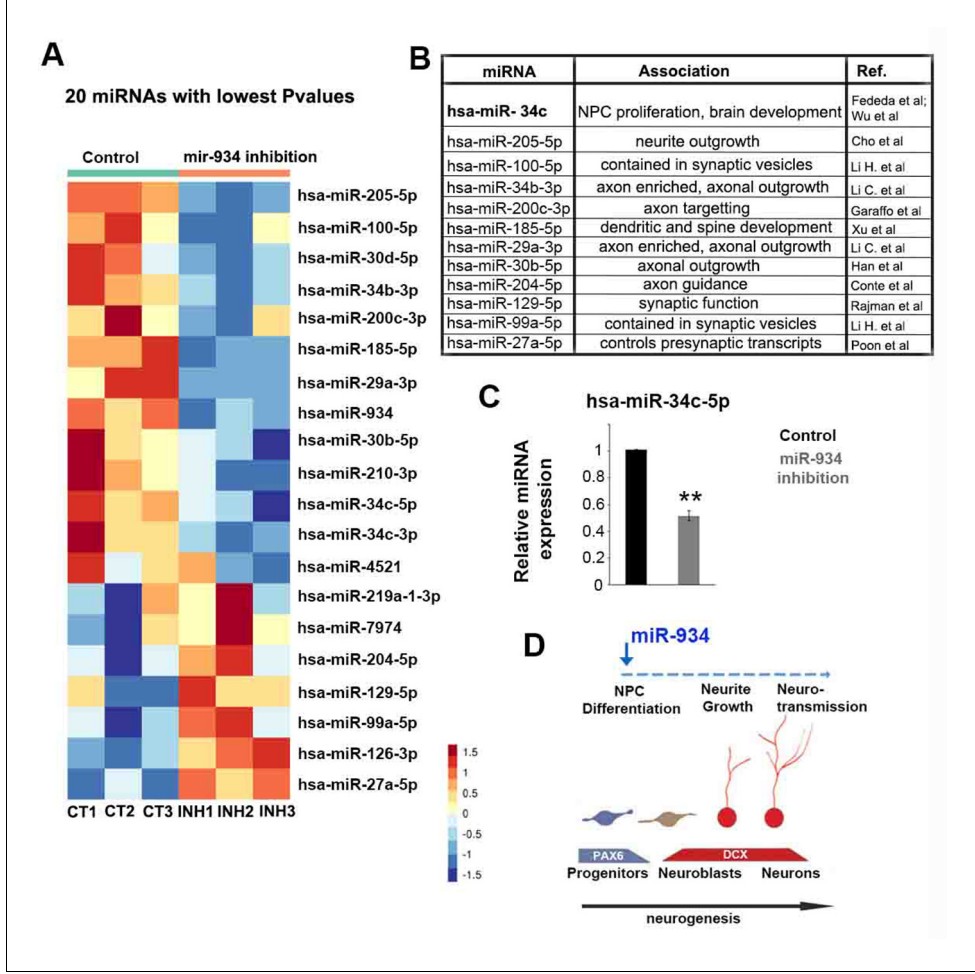

**Figure 6.** miRNA expression profiling following sustained inhibition of miR-934. (**A**) Heatmap showing the 20 miRNAs with lowest p-values at differential expression analysis following miR-934 sustained inhibition. (**B**) Table describing the association of differentially expressed miRNAs with axon/dendrite development and/or synapse formation. miRNA associations taken from *Fededa et al., 2016*; *Wu et al., 2014*; *Cho et al., 2013*; *Conte et al., 2014*; *Garaffo et al., 2015*; *Han et al., 2015*; *Li et al., 2019*; *Li et al., 2015*; *Poon et al., 2016*; *Rajman et al., 2017*; *Xu et al., 2013*. (**C**) qRT-PCR showing downregulation of mir34c-5p following miR-934 sustained inhibition (scrambled control 1±0.005 vs miR934 inhibition 0.51±0.03, n = 3, p=0.005). Bars and error bars represent mean values and the corresponding SEMs; 0.001<**p<0.01. (**D**) Sketch depicting the neurogenic function of miR-934 also affecting subsequent neuronal differentiation processes as reflected in large-scale changes in the expression of genes and miRNAs associated with neurogenesis.

The online version of this article includes the following source data for figure 6:

**Source data 1.** qRT-PCR data for expression of miR-34c-5p following miR-934 sustained inhibition.

brain development (*Fededa et al., 2016*; *Wu et al., 2014*), as a crucial regulator downstream of miR-934, and confirmed by RT-qPCR its reduction in response to miR-934 inhibition (*Figure 6c*; p=0.004. n = 4).

This data highlights the role of miR-934 in modulating molecular pathways mediating neurogenic events associated with early dorsal progenitor populations and the subplate zone.

## Discussion

Identification of species-specific features is critical in order to delineate spatiotemporal events during neural development, a process that underlies the complexity in organization and function of the human brain, and which may further convey the origin of neurodevelopmental or neuropsychiatric

pathologies. Using an in vitro model of neural development, we identified here miR-934 as a novel species-specific regulator of early human neurogenesis. miR-934 displays a developmental stage-specific expression pattern during neural induction that is characterized by neural progenitor expansion and early neuron generation. Our functional analysis revealed that it acts at the precursor-to-neuron transition to induce the differentiation of neural progenitors. The neurogenic function of miR-934 was further confirmed by large-scale changes in the expression of protein-coding and miRNA genes associated with progenitor cell proliferation and differentiation, neuritic growth and neurotransmission. The involvement of the identified miR-934 targets in these processes suggests a direct link between miR-934 function and the signalling networks mediating the neuronal differentiation program (*Figure 6d*).

Here we report four miR-934 targets during neural induction, namely TFCP2L1, FZD5, STMN2 and RAB3B. The direct downstream implications of miR-934 on post-transcriptional control of gene expression may be even broader since miRNAs, besides causing mRNA degradation, may also act to inhibit translation of their mRNA targets thus affecting expression at the protein level. The identified targets of miR-934 play a role in diverse processes along the neurogenic program including stem cell self-renewal, neural progenitor proliferation and differentiation as well as cytoskeletal arrangement and neurotransmission. It is therefore conceivable that miR-934-mediated modulation in the expression of these molecules should further influence downstream pathways related to morphological and functional transition of neural progenitor cells, thus causing an amplification in gene expression changes. Consistently, we observed altered expression in a total of 1458 genes upon miR-934 inhibition.

We present mechanistic and functional evidence that during neural induction miR-934 targets the Wnt receprtor FZD5 to suppress active β-catenin. Genetic ablation of FZD5 has previously been associated with increased generation of early-born retinal neurons and accelerated neurogenesis in the mouse (*Liu et al., 2012*). In line, β-catenin signaling hinders neuronal differentiation to promote progenitor cell proliferation during neurogenesis (*Woodhead et al., 2006*), while recently the Wnt/β-catenin pathway has been implicated in acquisition of early neural regional identities (*Yao et al., 2017*). Transcriptome profiling further supported the role of miR-934 in progenitor function by revealing altered expression of molecules that favor progenitor proliferation including N-cadherin, which liaises with β-catenin at adherens junctions to maintain NPC morphology and self-renewal (*Zhang et al., 2010*).

Differential gene expression analysis upon miR-934 inhibition revealed the microtubule destabilizing protein Stathmin 2, a powerful modulator of microtubule dynamics as a target of miR-934 (*Grenningloh et al., 2004*). The dynamic assembly and disassembly of microtubule cytoskeleton can dramatically impact on neurite/axonal outgrowth and placement of neurons (*Gärtner et al., 2014*), suggesting that the dysregulated expression of STMN2 contributes to the aberrant neuronal morphology noted here upon perturbations in miR-934 expression. In support, enrichment analysis further revealed negative regulation of cell adhesion and highlighted the involvement of permissive and repulsive cues required for axon–dendrite orientation, proper tract formation and neuronal positioning. These include semaphorins, their receptors neuropilins and plexins, and also Ephrins which are all components of the dynamic intrinsic–extrinsic process affecting axon outgrowth and guidance, and the bipolar transition required for radial migration of neurons during early development (*Polleux, 1998*; *Chen et al., 2008*; *Dimidschstein et al., 2013*; *Torii et al., 2009*).

In order for migrating neurons to respond to neurotransmitter-mediated signaling, specific organization of the neuronal cytoskeleton is required to permit selective transport of synaptic vesicles and neurotransmitter receptors to axons or dendrites (*Hirokawa et al., 2010*). Thus, the effect of miR-934 on neurotransmitter release may be considered as a consequence of abnormal neuritogenesis/axonogenesis and impaired microtubule dynamics. Interestingly, Rab3B, another identified target for miR-934 in this system, acts as a modulator of the neurotransmitter release machinery (*Schlüter et al., 2004*), while its overexpression results in inhibition of neurotransmitter secretion (*Schlüter et al., 2002*). Accordingly, perturbation of miR-934 results in altered expression of proteins implicated in the tight coupling of exo-endocytosis of synaptic vesicles, including Snapin and members of the synaptotagmin protein family (Syt11, Syt7) (*Chapman, 2002*; *Maximov et al., 2008*; *Wang et al., 2016*).

An intriguing finding is the upregulation of SYNGAP1 in response to miR-934 inhibition. SYN-GAP1 is localized in excitatory neurons (*Kim et al., 2003*) and acts as an essential repressive factor

to control the timing of dendritic spine synapse maturation (*Rumbaugh et al., 2006*; *Vazquez et al., 2004*). Moreover, SYNGAP1 has been associated with intellectual disability and autism spectrum disorders as its disruption affects the excitation/inhibition balance in neural networks that support cognition and behavior. Thus miR-934-mediated regulation of SYNGAP1 insinuates a link between early developmental events and shaping of connectivity at later stages, including integration of excitatory and inhibitory input (*La Fata et al., 2014*).

Consistent with the timing of miR-934 expression, positioned at an early neurodevelopmental period, we noted alterations in the expression of genes associated with the subplate, a region containing some of the earliest born cortical neurons (reviewed in *Luhmann et al., 2018*). During early development the structure and function of the cerebral cortex seems to be critically organized by SP neurons, a mostly transient heterogeneous population located below the cortical plate. SP neurons are indispensable during a restricted phase of development, contributing to refinement of the thalamocortical innervation, while it has been reported that the SP has the capacity to modulate projection neuron fate (*Ozair et al., 2018*). Already at GW9–GW10 in humans, the upper half of this compartment expresses markers that are characteristic of synaptic connectivity and axonal outgrowth (*Hoerder-Suabedissen and Molnár, 2015*).

In view of our collective functional and molecular data, miR-934 emerges as a regulator of early human neurogenesis instigating key transcriptional networks controlling progenitor cell proliferation and neuronal birth and differentiation, modulating the molecular profile of the SP zone. Our data are further complemented by differential expression of a distinct group of miRNAs also involved in neural progenitor proliferation or axon/neurite development and synapse formation. Further research on the function of species-specific miRNAs will advance our understanding on the mechanisms that control spatiotemporal events during human neurodevelopment, a process that dictates neuronal subtype specification, localization, and connectivity securing the integrity of higher cognitive functions.

# Materials and methods

**Key resources table**

| Reagent type (species) or resource | Designation | Source or reference | Identifiers | Additional information |
|---|---|---|---|---|
| Cell line (*H. sapiens*) | hESC line HUES6 | Cowan CA, Klimanskaya I, McMahon J, Atienza J, Witmyer J, Zucker JP, et al. Derivation of embryonic stem-cell lines from human blastocysts. The New England journal of medicine. 2004;350 (*Muñoz-Sanjuán and Brivanlou, 2002*):1353–6. | RRID:CVCL_B19 | Cell line (*H. sapiens*) |
| Chemical compound, drug | Knockout Serum Replacement (KSR) | Life Technologies Thermo Fisher | Life Technologies, Thermo Fisher 10828–010 | |
| Chemical compound, drug | KnockOut DMEM medium | Life Technologies Thermo Fisher | Life Technologies, Thermo Fisher 10829–018 | |
| Chemical compound, drug | matrigel matrix | BD biosciences | BD 354234 | See Materials and methods |
| Chemical compound, drug | Noggin | R and D Systems | R and D 6057 NG-25 | See Materials and methods |
| Chemical compound, drug | SB431542 | Tocris Biosciences | Tocris Biosciences 1614 | See Materials and methods |
| Commercial assay or kit | RNA Clean and Concentrator kit | Zymo Research | Zymo Research R1018 | |

*Continued on next page*

*Continued*

| Reagent type (species) or resource | Designation | Source or reference | Identifiers | Additional information |
|---|---|---|---|---|
| Recombinant DNA reagent | pmiR-GLO reporter vector | Promega | Promega E1330 | |
| Recombinant DNA reagent | miRZip lentivector-based anti-micro RNA system | Systems Biosciences | | https://systembio.com/products/mirna-and-lncrna-research-tools/mirzip-knockdown-vectors/mirzip-anti-mirna-constructs/ |
| Chemical compound, drug | Lipofectamine 2000 | Life Technologies/Thermo | Life Technologies 11668–019 | |
| Sequenced-based reagent | miR-934 mimics | Exiqon/Qiagen | | http://www.exiqon.com/ls/Pages/ProductDetails.aspx?ProductId=k1WVIuhVVQH848C13pCVyN5WGs82ItaE%2flSNAGCdK1%2bEWkr83WchdvRAbgzMNpIk3WDIx9O1eBJZUcYGe81LzgTaw2jJcGjXgRYNy58buck%3d&CategoryType=2yRAQGj8eprRK3%2bx8lE0zg%3d%3d |
| Sequenced-based reagent | miR-934 inhibitors | Dharmacon | | https://horizondiscovery.com/products/gene-modulation/knockdown-reagents/mirna/PIFs/miRIDIAN-microRNA-Hairpin-Inhibitor?nodeid=mirnamature-mimat0004977 |
| Antibody | mouse monoclonal anti-PAX6 | Developmental Studies Hybridoma Bank | RRID:AB_528427 | Dilution (1:50) See Materials and methods |
| Antibody | goat polyclonal anti-doublecortin (DCX) | Santa Cruz | Sc-8067 | Dilution (1:100) See Materials and methods |
| Antibody | rabbit polyclonal anti-Frizzled 5 | Abcam | Ab75234 | Dilution (1:1000) See Materials and methods |
| Antibody | rabbit monoclonal anti-active β-catenin | Cell signaling | Cell signaling, 19807 | Dilution (1:1000) See Materials and methods |
| Software, algorithm | ImageJ | ImageJ (http://imagej.nih.gov/ij/) | RRID:SCR_003070 | |
| Sequenced-based reagent | RT-qPCR primers | This paper | | See *Supplementary file 7* |

## hESC and iPSC culture

We have used the hESC line HUES6, which has been tested and shown free of mycoplasma contamination. The hESC line HUES6 (46XX) was kindly provided by Douglas Melton at Harvard University (*Cowan et al., 2004*); iPSC lines [C1-1 and C1-2 clones derived from skin fibroblasts of a male healthy donor and PD1-1 and PD1-2 clones derived from skin fibroblasts of a male patient with familial Parkinson's disease (PD) carrying the G209A (p.A53T) mutation in the α-synuclein gene SNCA] were generated and characterized as previously described (*Kouroupi et al., 2017*). HUES6/iPSC lines were cultured on irradiated mouse embryonic fibroblasts (MEFs, Globalstem) in HUES/iPSC medium consisting of KnockOut DMEM (KO-DMEM, Life Technologies), 20% Knockout Serum Replacement (KSR, Life Technologies), 2 mM GlutaMax, MEM Non-Essential Amino Acids (100x MEM NEAA, Life Technologies), 100 mM β-mercaptoethanol (Life Technologies) and 10 ng/ml human basic FGF2 (bFGF2, Miltenyi).

## Neural induction of hESCs/iPSCs

A schematic summary of the differentiation procedure is shown in *Figure 1a*. Cultures were disaggregated using accutase (Life Technologies) for 10 min at 37°C, collected in hESC/iPSC medium supplemented with 10 μM ROCK inhibitor Y-27632 (Tocris Bioscience) in the absence of bFGF-2 (embryoid body/EB medium) and replated in 60 mm petri dishes to allow embryoid body formation for 5–6 days. For neural induction, floating embryoid bodies were collected by centrifugation,

dissociated to single cells with accutase for 20 min at 37°C, and plated at a density of 50,000–65,000 cells/cm$^2$ on Matrigel (BD)-coated glass coverslips in a 24-well plate. Neural induction was achieved by dual suppression of the SMAD signaling pathway using a combination of Noggin (250 ng/ml; R and D Systems) and SB431542 (10 μM; Tocris Bioscience) in 1:1 mixture of Dulbecco's modified Eagle's medium with high glucose content (DMEM) and F-12, containing HEPES buffer (8 mM; Invitrogen), N2 supplement, and Glutamax (Gibco).

## Neuronal differentiation protocol

For neuronal differentiation, NPCs generated during the neural induction phase were dissociated with accutase and plated onto poly-L-ornithine (PLO; 20 μg/ml; Sigma-Aldrich)/laminin (5 μg/ml; Sigma-Aldrich)-coated coverslips. Neuronal differentiation was initiated for 8 days in Neurobasal medium supplemented with ascorbic acid (AA, Sigma-Aldrich; 200 μM), human recombinant sonic hedgehog (SHH, R and D Systems; 200 ng/ml), recombinant fibroblast growth factor 2b (FGF-2b, R and D Systems; 10 ng/ml), recombinant fibroblast growth factor 8b (FGF-8b, R and D Systems; 100 ng/ml), containing also 1% N2 supplement, 2% B-27 supplement and 1% Glutamax, followed by a cocktail consisting of brain-derived neurotrophic factor (BDNF, R and D Systems; 20 ng/ml), ascorbic acid (200 μM), glial cell-derived neurotrophic factor (GDNF, R and D Systems; 10 ng/ml), and cyclic-AMP (cAMP, Sigma-Aldrich; 0.5 mM) for further 21–26 days to allow neuronal maturation.

## RNA isolation, cDNA Synthesis and RT-qPCR

Total RNA, including miRNAs, was extracted from cell pellets at different time points of neural induction, using TRIzol Reagent (Life Technologies). Residual DNA was degraded with DNase (Takara) for 30 min at 37°C and purified RNA was recovered using the RNA Clean and Concentrator kit (Zymo Research). 1 μg of total RNA was used for first-strand cDNA synthesis which was performed using the PrimeScript RT reagent Kit (Takara). miRNA first-strand cDNA synthesis (from 10 ng RNA) was performed using the Universal cDNA synthesis kit (Exiqon). In all cases, commercially available kits were used according to manufacturer's instructions. Quantitative reverse transcription polymerase chain reaction (RT–qPCR) was carried out on a ViiA-7 real time PCR instrument (Applied Biosystems) according to the manufacturer's instructions. For mRNA expression detection, the RT–qPCR reactions were prepared with SYBR Green PCR Master mix (Kapa biosystems). LNA primer sets designed by Exiqon were used for miRNA expression detection and the RT–qPCR reactions were prepared with ExiLENT SYBR Green master mix (Exiqon). GAPDH and hsa-miR-103a-3p were used as reference genes. All primers used are listed in *Supplementary file 7*.

## Generation of genome-wide RNA sequencing data

RNA sequencing was performed at specific stages of hESC/iPSC directed differentiation and also upon sustained inhibition of miR-934 during neural induction of hESCs (NPC stage). In the first series of experiments, total cellular RNA isolated from samples at specific stages of the differentiation procedure (*Figure 1a*) corresponded to hESCs, hESC-derived NPCs (13 DIV) and neurons (48 DIV) alongside iPSCs, iPSC-derived NPCs and neurons cultured for the same DIV. Total RNA from two control iPSC lines (C1-1 and C1-2) derived from a healthy donor and two lines (PD1-1 and PD1-2) from a PD patient (*Kouroupi et al., 2017*). The originating fibroblasts as well as human fetal fibroblasts (Human Dermal Fibroblasts, foreskin, fetal; Cell Applications) were analyzed. In the second series, total RNA was isolated from three independent experiments of hESC-derived NPCs (15 DIV) transduced with an anti-miR-934 lentivirus or the scrambled control virus at DIV 10 of neural induction and collected 5 days later. Libraries of RNA transcripts following polyA selection were prepared using TruSeq RNA Sample Preparation Kit V2 (Illumina). Small RNA libraries were prepared using NEBNext Multiplex Small RNA Library Prep Set for Illumina. Next generation sequencing of RNAs (50 bp, paired-end) and small-RNAs (50 bp, single-end) was performed using an llumina HiSeq 2500 sequencer at the European Molecular Biology Laboratory (EMBL).

## Quantification and integration of expressed microRNAs and mRNAs

More than 2 billion reads were obtained from the performed sequencing experiments. All sequencing libraries were quality checked using FastQC (www.bioinformatics.babraham.ac.uk/projects/fastqc/). Contaminant detection and pre-processing was performed using a combination of in-house

scripts and available tools (*Vlachos et al., 2016*). RNA-Seq reads from the miR-934 activity inhibition experiments were aligned against GRCh38 human genome using STAR (*Dobin et al., 2013*) and quantified with featureCounts (*Liao et al., 2014*) against Ensembl v89. microRNA expression was quantified using miRDeep2 (*Mackowiak, 2011*). A gene/miRNA expression model encompassing the independent experiment number (i.e. 1, 2, 3) as a factor was built using DESeq2 (*Love et al., 2014*). RNA-Seq from the differentiation stages were analyzed as previously (*Kouroupi et al., 2017*). Differential expression analysis of miRNAs and RNAs across differentiation stages was performed using DESeq package in R (*Anders and Huber, 2010*). Functional analyses were performed using ClusterProfiler package (*Yu et al., 2012*). miR-934 expression across tissues was obtained by analyzing in-house 58 small RNA-Seq libraries from NCBI SRA repository with the same pipeline mentioned above for consistency. The accession IDs from the specific samples are indexed in *Supplementary file 4*. Validation of the differential expression for selected genes and miRNAs from the RNA-Seq analysis was performed by qRT-PCR in independent samples.

In silico miRNA target prediction was performed using DIANA-microT-CDS (*Paraskevopoulou et al., 2013*), while the detection of pathways regulated by differentially expressed miRNAs was realized using DIANA-miRPath v2.1 (*Vlachos et al., 2012*). Integration of small RNA-Seq and RNA-Seq expression data for the detection of miRNAs with central regulatory roles was performed with a local deployment of DIANA-mirExTra v2 (*Vlachos et al., 2016*). The following settings were utilized in the analysis: miRNA/RNA DE threshold: FDR < 0.05, microT-CDS prediction threshold: 0.7, DE method: DESeq.

In heatmaps, Pearson's correlation coefficient was used as a distance metric, while in the correlation plots the Euclidian distance between Pearson's correlation coefficients was employed for clustering. Expression Heatmap data were centered and scaled using row Z-scores. Fold-Change heatmaps were not centered/scaled or clustered; the column order follows the independent experiment number.

## Calculation of tissue-specificity-index tau for miRNAs across distinct differentiation stages

Tau was calculated on log2 (reads per million) values according to the recommended transformation (*Kryuchkova-Mostacci and Robinson-Rechavi, 2017*). miRNAs with tau >0.7 were selected as having highly specific expression signatures (n = 144), that is exhibiting disproportionally high expression in a single or few stages, while low or non-detectable in most other stages. The 144 miRNAs were subsequently ranked for median expression in NPCs in order to locate miRNAs with high and specific expression at the NPC stage.

## 3'-UTR dual luciferase assay

For cloning the 29nt long Fzd5 8-mer MRE, commercially available sense and anti-sense oligos (Eurofin) were annealed and inserted into the 3'-UTR of the pmiR-GLO reporter vector at the NheI and XbaI restriction sites. Oligos used for cloning are listed in *Supplementary file 7*. Clones were verified by sequencing analysis. HEK293T cells were co-transfected with the modified constructs and miRNA mimics (100 nM, Exiqon/Qiagen) and luciferase activity was measured 48 hr later with the Promega Luciferase assay system.

## Transduction with miRZip lentivirus for sustained miR-934 inhibition

Sustained inhibition of miR-934 during neural induction was achieved with the miRZip lentivector-based anti-microRNA system (Systems Biosciences). hESC cultures were transduced with the anti-miR-934 lentivirus or the scrambled control virus at DIV10 of neural induction and were collected for analysis 5 days later at DIV15. Samples were then examined by RNA sequencing, qRT-PCR or wWestern blot.

## Transfection with miRNA mimics or inhibitors

hESC cultures at neural induction stage were transiently transfected with Lipofectamine 2000 (Life Technology) in Optimem (ThermoFisher Scientific) according to the manufacturer's instructions. Cells were incubated in transfection mix overnight before fresh medium was added. A scrambled miRNA, which bears no homology to any known miRNA sequences in human, mouse or rat, was used as

control. In a first set of experiments, 3 doses of miR-934 mimics (100 nM, Exiqon) or inhibitors (100 nM, Dharmacon) were applied at DIV6, DIV8 and DIV10 and the cultures were analyzed 2 days later at DIV12. Cells were either fixed or collected with Accutase and the cultures were analyzed by immunocytochemistry and/or RT-qPCR.

## Immunocytochemistry and confocal microscopy

Cultures were fixed at the indicated time points in 4% w/v paraforlmadehyde for 20 min, washed with phosphate-buffered saline (PBS), blocked and permeabilized with 5% donkey serum in PBS containing 0.1% Triton X-100. The following primary antibodies were used: mouse monoclonal anti-PAX6 (1:50, Developmental Studies Hybridoma Bank) to identify neural progenitors; rabbit polyclonal anti-Nestin (1:200, Millipore) to identify neuroepithelial cells; goat polyclonal anti-doublecortin (DCX, 1:100; Santa Cruz) to identify neuroblasts/early neurons; rabbit polyclonal βIII-tubulin (TUJ-1 antibody 1:1000, Cell Signaling) to identify neurons; and goat polyclonal anti-Nanog (1:100, R and D) to identify pluripotent cells. After incubation with appropriate secondary antibodies conjugated with AlexaFluor 488 (green) or 546 (red) (1:500; Molecular Probes, Eugene, OR, USA), coverslips were mounted with ProLong Gold antifade reagent with DAPI (Cell Signaling). For quantification of labeled cells, images from 10 randomly selected fields from each independent experiment were obtained using a 20X lens on a Leica TCS-SP5II confocal microscope (LEICA Microsystems) and analyzed using ImageJ software (NIH).

## Western blot analysis

Cells were harvested, lysed at 4°C for 30 min in ice cold RIPA buffer containing PhosSTOP phosphatase inhibitors and a complete protease inhibitor mixture (Roche Life Science), and centrifuged at 20,000 g for 15 min. Protein concentration in the supernatant was estimated using the Bradford assay (Applichem). Proteins (50 µg) were separated by 10% SDS–PAGE electrophoresis and transferred onto nitrocellulose membranes (Trans-Blot, Biorad). Non-specific binding sites were blocked in Tris-buffered saline (TBS/0.1% Tween-20) containing 5% skimmed milk for 30 min at 37°C. Membranes were probed with the following primary antibodies: rabbit polyclonal anti-Frizzled 5 (1:1000, abcam), rabbit monoclonal anti-active β-catenin (1:1000, cell signaling), anti-Actin (1:1000, Millipore), and mouse monoclonal anti-GAPDH (1:1000, Santa Cruz). Immunodetection was performed by chemiluminescence (ECL) using the Amersham ECL Prime Western Blotting Detection Reagent (GE Healthcare) and quantification was performed against GAPDH and also verified against Actin.

## Morphological analysis of TUJ1+/DCX+ neurons

The outgrowth of the longest TUJ1+ process, and the percentage of neurons exhibiting a bipolar phenotype were estimated for TUJ+/DCX+ neurons in neural induction cultures transfected with miR-934 inhibitors. 160–280 neurons from four independent experiments were analyzed per condition using the Image J software and the Neuron J tool.

## Statistical analysis for low throughput experiments

All data were calculated as average of at least three independent experiments. Statistical analysis was performed using Student's t-test. All data are presented as the mean and standard error of the mean (mean ±SED). Significance is defined at ***p<0.001, **p<0.01, *p<0.05.

Details of the following Methods are presented in Appendix 1: In silico analysis of miR-934 species conservation and dual luciferase assays for identification of the binding site of miR934 on the 3'-UTR of FZD5, TFCP2L1 and RAB3B.

## Acknowledgements

This work was supported by Fondation Santé and the Greek General Secretariat for Research and Technology grants: EXCELLENCE 2272, and MIS 5002486 which is implemented under the 'Action for the Strategic Development on the Research and Technological Sector', funded by the Operational Programme 'Competitiveness, Entrepreneurship and Innovation' (NSRF 2014–2020) and co-financed by Greece and the European Union (European Regional Development Fund). This work was further supported by a Stavros Niarchos Foundation grant to the Hellenic Pasteur Institute as part of

the Foundation's initiative to support the Greek Research Center ecosystem and the Hellenic Foundation for Research and Innovation 899-PARKINSynapse grant to G.K.

## Additional information

### Funding

| Funder | Grant reference number | Author |
|---|---|---|
| Ministry of Education and Religious Affairs, Sport and Culture | Greek General Secreteriat for Research and Technology Grant EXCELLENCE 2272 | Rebecca Matsas |
| Ministry of Education and Religious Affairs, Sport and Culture | Greek General Secreteriat for Research and Technology Grant MIS 5002486 | Rebecca Matsas |
| Stavros Niarchos Foundation | StavrosNiarchos Foundation grant to the Hellenic Pasteur Institute as part of theFoundation's initiative to support the Greek Research Center ecosystem | Rebecca Matsas |
| The Hellenic Foundation for Research and Innovation | 899-PARKINSynapse grant | Georgia Kouroupi |
| Fondation Santé | An integrated genome-wide miRNA-mRNA approach in a human stem cell based model of neurodevelopment and disease | Rebecca Matsas |

The funders had no role in study design, data collection and interpretation, or the decision to submit the work for publication.

### Author contributions

Kanella Prodromidou, Conceptualization, Data curation, Investigation, Methodology, Conceived and designed the experiments, Performed the experiments and analyzed the data, Wrote the paper; Ioannis S Vlachos, Investigation, Methodology, Designed and performed the bioinformatics analysis, Provided input during writing; Maria Gaitanou, Georgia Kouroupi, Performed experiments; Artemis G Hatzigeorgiou, Supervision, Methodology, Supervised the bioinformatics analysis; Rebecca Matsas, Conceptualization, Data curation, Supervision, Funding acquisition, Investigation, Methodology, Conceived and designed the experiments, analyzed the data and wrote the paper

### Author ORCIDs

Kanella Prodromidou  https://orcid.org/0000-0001-5721-9124
Rebecca Matsas  https://orcid.org/0000-0002-4027-348X

### Ethics

Human subjects: All procedures for generation of human iPSCs were approved by the Scientific Council and Ethics Committee of Attikon University Hospital (Athens, Greece), which is one of the Mendelian forms of Parkinson's Disease clinical centers, and by the Hellenic Pasteur Institute Ethics Committee overlooking stem cell research. Informed consent was obtained from all donors before skin biopsy. Proc Natl Acad Sci U S A. 2017 May 2;114(18).

### Decision letter and Author response

Decision letter https://doi.org/10.7554/eLife.50561.sa1
Author response https://doi.org/10.7554/eLife.50561.sa2

# Additional files

## Supplementary files

• Supplementary file 1. Supplemental table 1 showing the 144 miRNAs across all differentiation stages demonstrating tissue specificity index tau >0.7.

• Supplementary file 2. Supplemental table 2 presenting The 10 most highly expressed miRNAs in NPCs demonstrating tissue specificity index tau >0.7.

• Supplementary file 3. Supplemental table 3 showing All Blastn results with evalue <1 for a query of the mature hsa-miR-934 sequence against all mature miRNAs in miRBase v22.1.

• Supplementary file 4. Supplemental table 4 presenting IDs and description of the small RNA-Seq human libraries of human tissue and cell data included in the analysis for miR934 expression.

• Supplementary file 5. Supplemental table 5 showing all the predicted mRNA targets for miRNA-934.

• Supplementary file 6. Supplemental table 6 presenting the 3' UTR binding sites for miR934 on its four identified targets (adapted from Diana microT-CDS).

• Supplementary file 7. Supplemental table 7 listing the sequences of primers and oligos used in this study.

• Supplementary file 8. Supplemental table 8 showing information on progenitor-specific and SP-enriched genes affected by sustained inhibition of miR-934.

• Transparent reporting form

## Data availability

Sequencing data have been deposited in GEO under accession code GSE101548. All data generated or analysed during this study are included in the manuscript and supporting files.

The following datasets were generated:

| Author(s) | Year | Dataset title | Dataset URL | Database and Identifier |
|---|---|---|---|---|
| Prodromidou K, Vlachos IS, Gaitanou M, Kouroupi G, Hatzigeorgiou AG, Matsas R | 2017 | MicroRNA-934 is a novel regulator orchestrating neurogenesis during early human neural development | https://www.ncbi.nlm.nih.gov/geo/query/acc.cgi?acc=GSE101548 | NCBI Gene Expression Omnibus, GSE101548 |
| Prodromidou K, Vlachos IS, Gaitanou M, Kouroupi G, Hatzigeorgiou AG, Matsas R | 2018 | MicroRNA-934 is a novel regulator orchestrating neurogenesis during early human neural development | https://www.ncbi.nlm.nih.gov/geo/query/acc.cgi?acc=GSE119760 | NCBI Gene Expression Omnibus, GSE119760 |

The following previously published datasets were used:

| Author(s) | Year | Dataset title | Dataset URL | Database and Identifier |
|---|---|---|---|---|
| Kuppusamya KT, Jones DC, Sperbera H, Madane A, Fischera KA, Rodriguez ML, Pabona L, Zhua W-Z, Tullocha NL, Yanga X, Sniadeckif NJ, Laflammea MA, Ruzzoc WL, Murrya CE, Ruohola-Bakera H | 2014 | Genome wide transcript and miRNAanalysis of invitro and in-vivo generated human cardiac samples | https://www.ncbi.nlm.nih.gov/sra/?term=SRR1636969 | NCBI Sequence Read Archive, SRR1636969 |
| Kuppusamya KT, Jones DC, Sperbera H, Madane A, Fischera KA, Rodriguez ML, Pabona L, | 2014 | Genome wide transcript and miRNAanalysis of invitro and in-vivo generated human cardiac samples | https://www.ncbi.nlm.nih.gov/sra/?term=SRR1636968 | NCBI Sequence Read Archive, SRR1636968 |

| | | | | |
|---|---|---|---|---|
| Zhua W-Z, Tullocha NL, Yanga X, Snia-deckif NJ, Laflam-mea MA, Ruzzoc WL, Murrya CE, Ruohola-Bakera H | | | | |
| Kuppusamya KT, Jones DC, Sper-bera H, Madane A, Fischera KA, Rodri-guez ML, Pabona L, Zhua W-Z, Tullocha NL, Yanga X, Snia-deckif NJ, Laflam-mea MA, Ruzzoc WL, Murrya CE, Ruohola-Bakera H | 2014 | Genome wide transcript and miRNAanalysis of invitro and in-vivo generated human cardiac samples | https://www.ncbi.nlm.nih.gov/sra/?term=SRR1636959 | NCBI Sequence Read Archive, SRR1636959 |
| Kuppusamya KT, Jones DC, Sper-bera H, Madane A, Fischera KA, Rodri-guez ML, Pabona L, Zhua W-Z, Tullocha NL, Yanga X, Snia-deckif NJ, Laflam-mea MA, Ruzzoc WL, Murrya CE, Ruohola-Bakera H | 2014 | Genome wide transcript and miRNAanalysis of invitro and in-vivo generated human cardiac samples | https://www.ncbi.nlm.nih.gov/sra/?term=SRR1636960 | NCBI Sequence Read Archive, SRR1636960 |
| Kuppusamya KT, Jones DC, Sper-bera H, Madane A, Fischera KA, Rodri-guez ML, Pabona L, Zhua W-Z, Tullocha NL, Yanga X, Snia-deckif NJ, Laflam-mea MA, Ruzzoc WL, Murrya CE, Ruohola-Bakera H | 2014 | Genome wide transcript and miRNAanalysis of invitro and in-vivo generated human cardiac samples | https://www.ncbi.nlm.nih.gov/sra/?term=SRR1636962 | NCBI Sequence Read Archive, SRR1636962 |
| Kuppusamya KT, Jones DC, Sper-bera H, Madane A, Fischera KA, Rodri-guez ML, Pabona L, Zhua W-Z, Tullocha NL, Yanga X, Snia-deckif NJ, Laflam-mea MA, Ruzzoc WL, Murrya CE, Ruohola-Bakera H | 2014 | Genome wide transcript and miRNAanalysis of invitro and in-vivo generated human cardiac samples | https://www.ncbi.nlm.nih.gov/sra/?term=SRR1636963 | NCBI Sequence Read Archive, SRR1636963 |
| Kuppusamya KT, Jones DC, Sper-bera H, Madane A, Fischera KA, Rodri-guez ML, Pabona L, Zhua W-Z, Tullocha NL, Yanga X, Snia-deckif NJ, Laflam-mea MA, Ruzzoc WL, Murrya CE, Ruohola-Bakera H | 2014 | Genome wide transcript and miRNAanalysis of invitro and in-vivo generated human cardiac samples | https://www.ncbi.nlm.nih.gov/sra/?term=SRR1636965 | NCBI Sequence Read Archive, SRR1636965 |
| Jönsson ME, Wah-lestedt JN, Åker-blom M, Kirkeby A, Malmevik J, Brat-taas PL, Jakobsson J, Parmar M | 2015 | Comprehensive analysis of microRNA expression in the human developing brain reveals microRNA-10 as a caudalizing factor | https://www.ncbi.nlm.nih.gov/sra/?term=SRR1988287 | NCBI Sequence Read Archive, SRR1988287 |
| Jönsson ME, Wah-lestedt JN, Åker-blom M, Kirkeby A, | 2015 | Comprehensive analysis of microRNA expression in the human developing brain reveals | https://www.ncbi.nlm.nih.gov/sra/?term=SRR1988288 | NCBI Sequence Read Archive, SRR1988288 |

| | | | | |
|---|---|---|---|---|
| Malmevik J, Brattaas PL, Jakobsson J, Parmar M | | microRNA-10 as a caudalizing factor | | |
| Jönsson ME, Wahlestedt JN, Åkerblom M, Kirkeby A, Malmevik J, Brattaas PL, Jakobsson J, Parmar M | 2015 | Comprehensive analysis of microRNA expression in the human developing brain reveals microRNA-10 as a caudalizing factor | https://www.ncbi.nlm.nih.gov/sra/?term=SRR1988291 | NCBI Sequence Read Archive, SRR1988291 |
| Jönsson ME, Wahlestedt JN, Åkerblom M, Kirkeby A, Malmevik J, Brattaas PL, Jakobsson J, Parmar M | 2015 | Comprehensive analysis of microRNA expression in the human developing brain reveals microRNA-10 as a caudalizing factor | https://www.ncbi.nlm.nih.gov/sra/?term=SRR1988292 | NCBI Sequence Read Archive, SRR1988292 |
| Santa-Maria I, Alaniz ME, Renwick N, Cela C, Fulga TA, Vactor DV, Tuschl T, Clark LN, Shelanski ML, McCabe BD, Crary JF | 2014 | Dysregulation of microRNAs in neurodegeneration | https://www.ncbi.nlm.nih.gov/sra/?term=SRR1658346 | NCBI Sequence Read Archive, SRR1658346 |
| Santa-Maria I, Alaniz ME, Renwick N, Cela C, Fulga TA, Vactor DV, Tuschl T, Clark LN, Shelanski ML, McCabe BD, Crary JF | 2014 | Dysregulation of microRNAs in neurodegeneration | https://www.ncbi.nlm.nih.gov/sra/?term=SRR1658360 | NCBI Sequence Read Archive, SRR1658360 |
| Hoss AG, Labadorf A, Latourelle JC, Kartha VK, Hadzi TC, Gusella JF, MacDonald ME, ChenJ-F, Akbarian S, Weng Z, Vonsattel JP, Myers RH | 2015 | miRNA-seq expression profiling of Huntington's Disease and neurologically normal human post-mortem prefrontal cortex (BA9) brain samples | https://www.ncbi.nlm.nih.gov/sra/?term=SRR1759212 | NCBI Sequence Read Archive, SRR1759212 |
| Hoss AG, Labadorf A, Latourelle JC, Kartha VK, Hadzi TC, Gusella JF, MacDonald ME, ChenJ-F, Akbarian S, Weng Z, Vonsattel JP, Myers RH | 2015 | miRNA-seq expression profiling of Huntington's Disease and neurologically normal human post-mortem prefrontal cortex (BA9) brain samples | https://www.ncbi.nlm.nih.gov/sra/?term=SRR1759213 | NCBI Sequence Read Archive, SRR1759213 |
| Lopez JP, Diallo A, Cruceanu C, Fiori LM, Laboissiere S, Guillet I, Fontaine J, Ragoussis J, Benes V, Turecki G, Ernst C | 2015 | Biomarker discovery: Quantification of microRNAs and other small non-coding RNAs using next generation sequencing | https://www.ncbi.nlm.nih.gov/sra/?term=SRR2061800 | NCBI Sequence Read Archive, SRR2061800 |
| Lopez JP, Diallo A, Cruceanu C, Fiori LM, Laboissiere S, Guillet I, Fontaine J, Ragoussis J, Benes V, Turecki G, Ernst C | 2015 | Biomarker discovery: Quantification of microRNAs and other small non-coding RNAs using next generation sequencing | https://www.ncbi.nlm.nih.gov/sra/?term=SRR2061801 | NCBI Sequence Read Archive, SRR2061801 |
| Lopez JP, Diallo A, Cruceanu C, Fiori LM, Laboissiere S, Guillet I, Fontaine J, Ragoussis J, Benes V, Turecki G, Ernst C | 2015 | Biomarker discovery: Quantification of microRNAs and other small non-coding RNAs using next generation sequencing | https://www.ncbi.nlm.nih.gov/sra/?term=SRR2061795 | NCBI Sequence Read Archive, SRR2061795 |
| Lopez JP, Diallo A, Cruceanu C, Fiori | 2015 | Biomarker discovery: Quantification of microRNAs and other small non- | https://www.ncbi.nlm.nih.gov/sra/?term= | NCBI Sequence Read Archive, SRR20 |

| | | | | | |
|---|---|---|---|---|---|
| LM, Laboissiere S, Guillet I, Fontaine J, Ragoussis J, Benes V, Turecki G, Ernst C | | | coding RNAs using next generation sequencing | SRR2061797 | 61797 |
| Lopez JP, Diallo A, Cruceanu C, Fiori LM, Laboissiere S, Guillet I, Fontaine J, Ragoussis J, Benes V, Turecki G, Ernst C | 2015 | Biomarker discovery: Quantification of microRNAs and other small non-coding RNAs using next generation sequencing | https://www.ncbi.nlm.nih.gov/sra/?term=SRR2061803 | NCBI Sequence Read Archive, SRR2061803 |
| Lopez JP, Diallo A, Cruceanu C, Fiori LM, Laboissiere S, Guillet I, Fontaine J, Ragoussis J, Benes V, Turecki G, Ernst C | 2015 | Biomarker discovery: Quantification of microRNAs and other small non-coding RNAs using next generation sequencing | https://www.ncbi.nlm.nih.gov/sra/?term=SRR2061804 | NCBI Sequence Read Archive, SRR2061804 |
| Lopez JP, Diallo A, Cruceanu C, Fiori LM, Laboissiere S, Guillet I, Fontaine J, Ragoussis J, Benes V, Turecki G, Ernst C | 2015 | Biomarker discovery: Quantification of microRNAs and other small non-coding RNAs using next generation sequencing | https://www.ncbi.nlm.nih.gov/sra/?term=SRR2061810 | NCBI Sequence Read Archive, SRR2061810 |
| Hébert SS, Wang W-X, Zhu Q, Nelson PT | 2013 | A Study of Small RNAs from Cerebral Neocortex of Pathology-Verified Alzheimer's Disease, Dementia with Lewy Bodies, Hippocampal Sclerosis, Frontotemporal Lobar Dementia, and Non-Demented Human Controls | https://www.ncbi.nlm.nih.gov/sra/?term=SRR828708 | NCBI Sequence Read Archive, SRR828708 |
| Hébert SS, Wang W-X, Zhu Q, Nelson PT | 2013 | A Study of Small RNAs from Cerebral Neocortex of Pathology-Verified Alzheimer's Disease, Dementia with Lewy Bodies, Hippocampal Sclerosis, Frontotemporal Lobar Dementia, and Non-Demented Human Controls | https://www.ncbi.nlm.nih.gov/sra/?term=SRR828709 | NCBI Sequence Read Archive, SRR828709 |
| Cheng W-C, Chung I-F, Huang T-S, Chang S-T, Sun H-J, Tsai C-F, Liang M-L, Wong T-T, Wang H-W | 2012 | YM500: An integrative small RNA sequencing (smRNA-seq) database for microRNA research | https://www.ncbi.nlm.nih.gov/sra/?term=SRR531688 | NCBI Sequence Read Archive, SRR531688 |
| Cheng W-C, Chung I-F, Huang T-S, Chang S-T, Sun H-J, Tsai C-F, Liang M-L, Wong T-T, Wang H-W | 2012 | YM500: An integrative small RNA sequencing (smRNA-seq) database for microRNA research | https://www.ncbi.nlm.nih.gov/sra/?term=SRR531687 | NCBI Sequence Read Archive, SRR531687 |
| Cheng W-C, Chung I-F, Huang T-S, Chang S-T, Sun H-J, Tsai C-F, Liang M-L, Wong T-T, Wang H-W | 2012 | YM500: An integrative small RNA sequencing (smRNA-seq) database for microRNA research | https://www.ncbi.nlm.nih.gov/sra/?term=SRR531692 | NCBI Sequence Read Archive, SRR531692 |
| Cheng W-C, Chung I-F, Huang T-S, Chang S-T, Sun H-J, Tsai C-F, Liang M-L, Wong T-T, Wang H-W | 2012 | YM500: An integrative small RNA sequencing (smRNA-seq) database for microRNA research | https://www.ncbi.nlm.nih.gov/sra/?term=SRR531694 | NCBI Sequence Read Archive, SRR531694 |
| Cheng W-C, Chung I-F, Huang T-S, | 2012 | YM500: An integrative small RNA sequencing (smRNA-seq) database | https://www.ncbi.nlm.nih.gov/sra/?term= | NCBI Sequence Read Archive, |

| Author | Year | Title | URL | Database |
|---|---|---|---|---|
| Chang S-T, Sun H-J, Tsai C-F, Liang M-L, Wong T-T, Wang H-W | | for microRNA research | SRR531683 | SRR531683 |
| Cheng W-C, Chung I-F, Huang T-S, Chang S-T, Sun H-J, Tsai C-F, Liang M-L, Wong T-T, Wang H-W | 2012 | YM500: An integrative small RNA sequencing (smRNA-seq) database for microRNA research | https://www.ncbi.nlm.nih.gov/sra/?term=SRR531684 | NCBI Sequence Read Archive, SRR531684 |
| Tabassum R, Sivadas A, Agrawal V, Tian H, Arafat D, Gibson G | 2015 | Omic Personality: Implications of Stable Transcript and Methylation Profiles for Personalized Medicine | https://www.ncbi.nlm.nih.gov/sra/?term=SRR1949839 | NCBI Sequence Read Archive, SRR1949839 |
| Tabassum R, Sivadas A, Agrawal V, Tian H, Arafat D, Gibson G | 2015 | Omic Personality: Implications of Stable Transcript and Methylation Profiles for Personalized Medicine | https://www.ncbi.nlm.nih.gov/sra/?term=SRR1949841 | NCBI Sequence Read Archive, SRR1949841 |
| Tabassum R, Sivadas A, Agrawal V, Tian H, Arafat D, Gibson G | 2015 | Omic Personality: Implications of Stable Transcript and Methylation Profiles for Personalized Medicine | https://www.ncbi.nlm.nih.gov/sra/?term=SRR1949847 | NCBI Sequence Read Archive, SRR1949847 |
| Tabassum R, Sivadas A, Agrawal V, Tian H, Arafat D, Gibson G | 2015 | Omic Personality: Implications of Stable Transcript and Methylation Profiles for Personalized Medicine | https://www.ncbi.nlm.nih.gov/sra/?term=SRR1949850 | NCBI Sequence Read Archive, SRR1949850 |
| Tabassum R, Sivadas A, Agrawal V, Tian H, Arafat D, Gibson G | 2015 | Omic Personality: Implications of Stable Transcript and Methylation Profiles for Personalized Medicine | https://www.ncbi.nlm.nih.gov/sra/?term=SRR1949858 | NCBI Sequence Read Archive, SRR1949858 |
| Tabassum R, Sivadas A, Agrawal V, Tian H, Arafat D, Gibson G | 2015 | Omic Personality: Implications of Stable Transcript and Methylation Profiles for Personalized Medicine | https://www.ncbi.nlm.nih.gov/sra/?term=SRR1949861 | NCBI Sequence Read Archive, SRR1949861 |
| McLean CS, Mielke C, Cordova JM, Langlais PR, Bowen B, Miranda D, Coletta DK, Mandarino LJ | 2015 | Gene and MicroRNA Expression Responses to Exercise; Relationship with Insulin Sensitivity | https://www.ncbi.nlm.nih.gov/sra/?term=SRR1820679 | NCBI Sequence Read Archive, (SRA) SRR1820679 |
| McLean CS, Mielke C, Cordova JM, Langlais PR, Bowen B, Miranda D, Coletta DK, Mandarino LJ | 2015 | Gene and MicroRNA Expression Responses to Exercise; Relationship with Insulin Sensitivity | https://www.ncbi.nlm.nih.gov/sra/?term=SRR1820680 | NCBI Sequence Read Archive, SRR1820680 |
| Gulati N, Løvendorf MB, Zibert JR, Akat KM, Renwick N, Tuschl T, Krueger JG | 2015 | Unique microRNAs appear at different times during the course of a delayed-type hypersensitivity reaction in human skin | https://www.ncbi.nlm.nih.gov/sra/?term=SRR2174513 | NCBI Sequence Read Archive, SRR2174513 |
| Gulati N, Løvendorf MB, Zibert JR, Akat KM, Renwick N, Tuschl T, Krueger JG | 2015 | Unique microRNAs appear at different times during the course of a delayed-type hypersensitivity reaction in human skin | https://www.ncbi.nlm.nih.gov/sra/?term=SRR2174514 | NCBI Sequence Read Archive, SRR2174514 |
| Gulati N, Løvendorf MB, Zibert JR, Akat KM, Renwick N, Tuschl T, Krueger JG | 2015 | Unique microRNAs appear at different times during the course of a delayed-type hypersensitivity reaction in human skin | https://www.ncbi.nlm.nih.gov/sra/?term=SRR2174515 | NCBI Sequence Read Archive, SRR2174515 |
| Gulati N, Løvendorf MB, Zibert JR, Akat KM, Renwick N, Tuschl T, Krueger JG | 2015 | Unique microRNAs appear at different times during the course of a delayed-type hypersensitivity reaction in human skin | https://www.ncbi.nlm.nih.gov/sra/?term=SRR2174516 | NCBI Sequence Read Archive, SRR2174516 |

| | | | | |
|---|---|---|---|---|
| Gulati N, Løvendorf MB, Zibert JR, Akat KM, Renwick N, Tuschl T, Krueger JG | 2015 | Unique microRNAs appear at different times during the course of a delayed-type hypersensitivity reaction in human skin | https://www.ncbi.nlm.nih.gov/sra/?term=SRR2174517 | NCBI Sequence Read Archive, SRR2174517 |
| Gulati N, Løvendorf MB, Zibert JR, Akat KM, Renwick N, Tuschl T, Krueger JG | 2015 | Unique microRNAs appear at different times during the course of a delayed-type hypersensitivity reaction in human skin | https://www.ncbi.nlm.nih.gov/sra/?term=SRR2174518 | NCBI Sequence Read Archive, SRR2174518 |
| Gulati N, Løvendorf MB, Zibert JR, Akat KM, Renwick N, Tuschl T, Krueger JG | 2015 | Unique microRNAs appear at different times during the course of a delayed-type hypersensitivity reaction in human skin | https://www.ncbi.nlm.nih.gov/sra/?term=SRR2174519 | NCBI Sequence Read Archive, SRR2174519 |
| Gulati N, Løvendorf MB, Zibert JR, Akat KM, Renwick N, Tuschl T, Krueger JG | 2015 | Unique microRNAs appear at different times during the course of a delayed-type hypersensitivity reaction in human skin | https://www.ncbi.nlm.nih.gov/sra/?term=SRR2174520 | NCBI Sequence Read Archive, SRR2174520 |
| Gulati N, Løvendorf MB, Zibert JR, Akat KM, Renwick N, Tuschl T, Krueger JG | 2015 | Unique microRNAs appear at different times during the course of a delayed-type hypersensitivity reaction in human skin | https://www.ncbi.nlm.nih.gov/sra/?term=SRR2174537 | NCBI Sequence Read Archive, SRR2174537 |
| Gulati N, Løvendorf MB, Zibert JR, Akat KM, Renwick N, Tuschl T, Krueger JG | 2015 | Unique microRNAs appear at different times during the course of a delayed-type hypersensitivity reaction in human skin | https://www.ncbi.nlm.nih.gov/sra/?term=SRR2174538 | NCBI Sequence Read Archive, SRR2174538 |
| Gulati N, Løvendorf MB, Zibert JR, Akat KM, Renwick N, Tuschl T, Krueger JG | 2015 | Unique microRNAs appear at different times during the course of a delayed-type hypersensitivity reaction in human skin | https://www.ncbi.nlm.nih.gov/sra/?term=SRR2174541 | NCBI Sequence Read Archive, SRR2174541 |
| Gulati N, Løvendorf MB, Zibert JR, Akat KM, Renwick N, Tuschl T, Krueger JG | 2015 | Unique microRNAs appear at different times during the course of a delayed-type hypersensitivity reaction in human skin | https://www.ncbi.nlm.nih.gov/sra/?term=SRR2174542 | NCBI Sequence Read Archive, SRR2174542 |
| Farazi TA, Horlings HM, Hoeve JT, Mihailovic A, Halfwerk H, Morozov P, Brown M, Hafner M, Reyal F, Kouwenhove MV, Kreike B, Sie D, Hovestadt V, Wessels L, de Vijver MJ, Tuschl T | 2011 | MicroRNA sequence and expression analysis in breast tumors by deep sequencing | https://www.ncbi.nlm.nih.gov/sra/?term=SRR191548 | NCBI Sequence Read Archive, SRR191548 |
| Farazi TA, Horlings HM, Hoeve JT, Mihailovic A, Halfwerk H, Morozov P, Brown M, Hafner M, Reyal F, Kouwenhove MV, Kreike B, Sie D, Hovestadt V, Wessels L, de Vijver MJ, Tuschl T | 2011 | MicroRNA sequence and expression analysis in breast tumors by deep sequencing | https://www.ncbi.nlm.nih.gov/sra/?term=SRR191578 | NCBI Sequence Read Archive, SRR191578 |
| Zhou L, Chen J, Li Z, Li X, Hu X, Huang X, Zhao X, Liang C, Wang Y, Sun L, Shi M, Xu X, Shen F, Chen M, Han Z, Peng Z, Zhai Q, Zhang Z, Yang | 2010 | miRNA sequencing of 10 pairs samples between kidney normal tissues and cancer tissue | https://www.ncbi.nlm.nih.gov/sra/?term=SRR070232 | NCBI Sequence Read Archive, SRR070232 |

| | | | | | |
|---|---|---|---|---|---|
| R, Ye J, Guan Z, Yang H, Gui Y, Wang J, Cai Z, Zhang X | | | | | |
| Zhou L, Chen J, Li Z, Li X, Hu X, Huang X, Zhao X, Liang C, Wang Y, Sun L, Shi M, Xu X, Shen F, Chen M, Han Z, Peng Z, Zhai Q, Zhang Z, Yang R, Ye J, Guan Z, Yang H, Gui Y, Wang J, Cai Z, Zhang X | 2010 | miRNA sequencing of 10 pairs samples between kidney normal tissues and cancer tissue | https://www.ncbi.nlm.nih.gov/sra/?term=SRR070230 | NCBI Sequence Read Archive, SRR070 230 |
| Nowakowski TJ, Rani N, Golkaram M, Zhou HR, Al-varadoB, Huch K, West JA, Leyrat A, Pollen AA, Kriegstein AR, Petzold LR, Kosik KS | 2017 | Dynamic microRNA regulatory networks in establishing the Cellular specificity of developing human brains | https://www.ncbi.nlm.nih.gov/sra/?term=SRR6328631 | NCBI Sequence Read Archive, SRR6328631 |
| Nowakowski TJ, Rani N, Golkaram M, Zhou HR, Al-varadoB, Huch K, West JA, Leyrat A, Pollen AA, Kriegstein AR, Petzold LR, Kosik KS | 2017 | Dynamic microRNA regulatory networks in establishing the Cellular specificity of developing human brains | https://www.ncbi.nlm.nih.gov/sra/?term=SRR6328627 | NCBI Sequence Read Archive, SRR6328627 |
| Nowakowski TJ, Rani N, Golkaram M, Zhou HR, Al-varadoB, Huch K, West JA, Leyrat A, Pollen AA, Kriegstein AR, Petzold LR, Kosik KS | 2017 | Dynamic microRNA regulatory networks in establishing the Cellular specificity of developing human brains | https://www.ncbi.nlm.nih.gov/sra/?term=SRR6328628 | NCBI Sequence Read Archive, SRR6328628 |
| Nowakowski TJ, Rani N, Golkaram M, Zhou HR, Al-varadoB, Huch K, West JA, Leyrat A, Pollen AA, Kriegstein AR, Petzold LR, Kosik KS | 2017 | Dynamic microRNA regulatory networks in establishing the Cellular specificity of developing human brains | https://www.ncbi.nlm.nih.gov/sra/?term=SRR6328630 | NCBI Sequence Read Archive, SRR6328630 |

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

## Appendix 1

## In silico analysis of miR-934 species conservation

To identify other species having potentially mature miRNAs with similar sequences we performed two distinct analyses: A) We searched using a sensitive blastn (The BLAST Sequence Analysis Tool, The NCBI Handbook [Internet]. Second Edition. Thomas Madden, Created March 15, 2013) query (e-value cutoff 1, word size 4, match score +5, mismatch penalty −4) all mature miRNAs annotated in miRBase v22.1 database (*Kozomara et al., 2019*) against the full hsa-miR-934 mature sequence. The dataset comprises 38,859 miRNAs across 271 species. In the latest version, additional support for miRNAs is provided by the analysis of 1493 small RNA-Seq datasets. (**B**) We searched using a sensitive blastn query with similar settings as above against all miRNAs (mature, precursors) identified in MirGeneDB2.0 (*Fromm et al., 2020*), a new resource aiming to independently annotate miRNA genes. The dataset comprises >10,000 miRNA genes from 45 organisms identified by the analysis of >400 small RNA-Seq datasets. Both queries identified only primate species having a mature miRNA or miRNA precursor with similarity (e-value <1) to the mature hsa-miR-934 sequence. miRBase database, since it is a richer resource, enabled us to identify four species (apart from human) where miR-934 is expressed (*Supplementary file 2*). The query against miRGeneDB 2.0 identified only hsa-miR-934 (mature and gene) as well as mml-miR-934 (mature and gene) as significant hits (e-value <1). Only primate species (*Homo sapiens*, Pan troglodytes, Pongo pygmaeus, Macaca mulatta, Callithrix jacchus) were detected by querying all sequences deposited in the two largest datasets of mature miRNAs and miRNA precursors, providing further evidence that miR-934 is a primate specific miRNA.

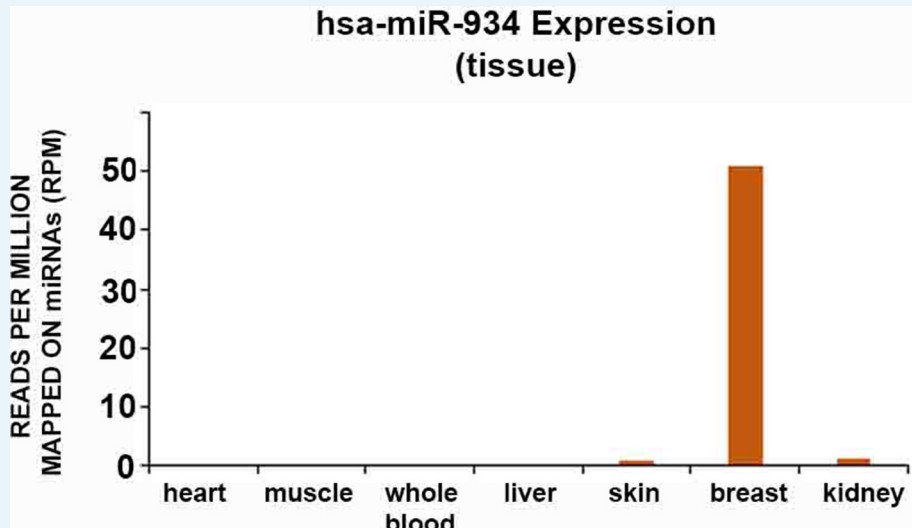

**Appendix 1—figure 1.** Graph incorporating miRNA expression data from uniformly-analyzed small RNA-Seq libraries of non-neural human tissue (detailed in *Supplementary file 3*). Association was noted only in one case (breast).

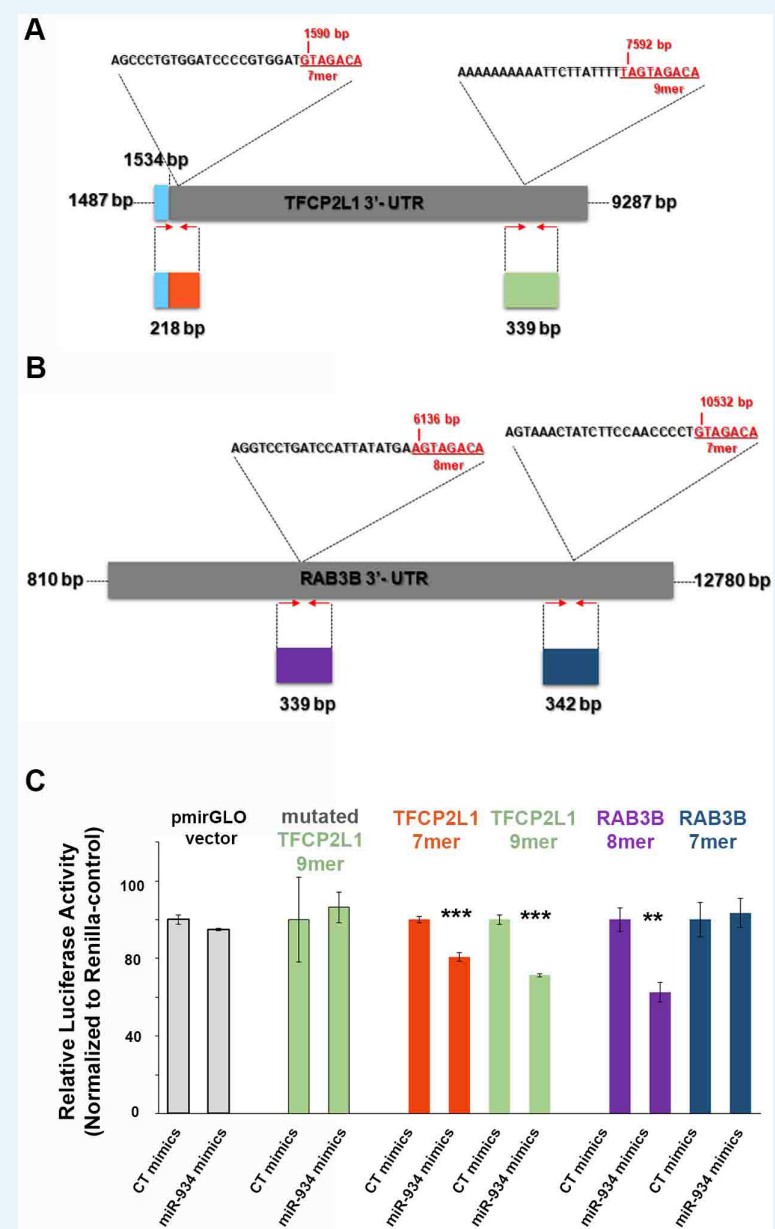

**Appendix 1—figure 2.** Identification of binding sites for miR-934 on the 3'-UTR of TFCP2L1 and RAB3B. (**A**) Schematic representation of TFCP2L1 3'-UTR depicting in red fonts the seed-binding sequences of the two predicted MREs for miR-934 with their respective positions. To examine mechanistically the inhibition exerted by miR-934 on TFCP2L1 expression, a luciferase reporter system was used to evaluate independently the binding of miR-934 on each of the 2 MREs which contain either a 7mer, or a 9-mer site. Shown in color code are the sequences and sizes of TFCP2L1 3'-UTR regions containing the 2 MREs, as they were cloned into the pmirGLO vector [orange: 7mer (blue box represents the last 47 base pairs at the 3' end of the human TFCP2L1 coding region); green: 9mer]. Arrows indicate primer sites for cloning. Fragments of the 3'-UTR of TFCP2L1 containing each of the two possible binding domains for miR-934 (218 bp, including the 7mer sequence; and 339 bp including the 9-mer sequence) were cloned separately into the 3'-UTR of the dual luciferase reporter construct pmirGLO. (**B**) Schematic representation of RAB3B 3'-UTR depicting in red fonts the seed-binding sequences of the two predicted MREs for miR-934 with their respective positions. To examine mechanistically the inhibition exerted by miR-934 on RAB3B expression, a luciferase reporter system was used to evaluate independently the binding of miR-934 on each of the 2 MREs,

which contain either a 8mer or a 7mer site. Shown in color code are the sequences and sizes of RAB3B 3'-UTR regions containing the 2 MREs, as they were cloned into the pmirGLO vector (purple: 8mer; dark blue: 7mer). Arrows indicate primer sites for cloning. Fragments of the 3'-UTR of RAB3B containing each of the two possible binding domains for miR-934 (339 bp, including the 8mer sequence; and 342 bp including the 7-mer sequence) were cloned separately into the 3'-UTR of the dual luciferase reporter construct pmirGLO. (C) Upon co-transfection of HEK293T cells with miR-934 mimics and the different TFCP2L1 and RAB3B reporter constructs, luciferase activity was suppressed in both constructs for TFCP2L1 (p=0.0002 for 7mer and p=0.0007 for 9mer) and only in the construct containing the 8mer site sequence for RAB3B (p=0.002). By contrast luciferase activity was not affected upon transfection of control constructs (pmiRGLO vector and the mutated TFCP2L1 9mer) in the presence of miR934 mimics. Color code as in (A) and (B).

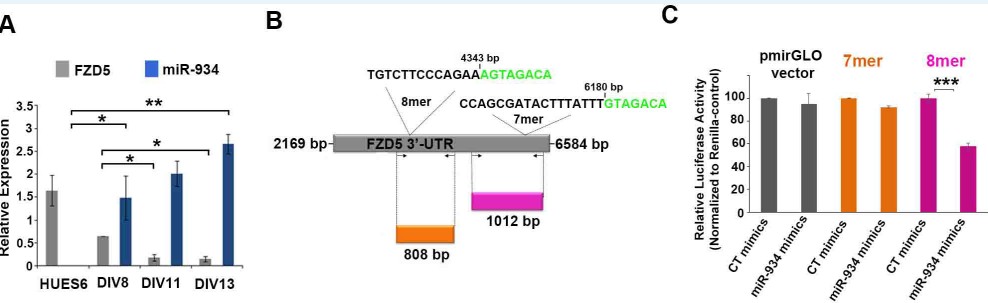

**Appendix 1—figure 3.** Identification of an 8mer-seed sequence as binding site for miR-934 on the 3'-UTR of Fzd5. (**A**) RT-qPCR analysis at different time points of neural induction, showing up-regulation of miR-934 and parallel down-regulation of the mRNA of its predicted target, Fzd5. Using the DIANA-microT-CDS target prediction algorithm, two putative MREs were identified on the 3'-UTR of Fzd5. (**B**) Schematic representation of Fzd5 3'-UTR depicting in green characters the seed-binding sequences of the two predicted MREs for miR-934 with their respective positions. To examine mechanistically the inhibition exerted by miR-934 on Fzd5 expression, a luciferase reporter system was used to evaluate independently the binding of miR-934 on each of the 2 MREs which contain either a 7mer (binding score: 0.032), or an 8-mer (binding score: 0.005) site. Shown in color code are the sequences and sizes of Fzd5 3'-UTR regions containing the 2 MREs, as they were cloned into the pmirGLO vector (orange: 7mer; magenta: 8mer). Arrows indicate primer sites for cloning. Fragments of the 3'-UTR of Fzd5 containing each of the two possible binding domains for miR-934 (1012 bp, including the 7mer sequence; and 808 bp including the 8-mer sequence) were cloned separately into the 3'-UTR of the dual luciferase reporter construct pmirGLO. (**C**) Upon co-transfection of HEK293T cells with miR-934 mimics and the different Fzd5 reporter constructs, luciferase activity was only suppressed in the construct (808 bp) containing the 8mer site sequence (p=0.000008). Color code as in (**A**).

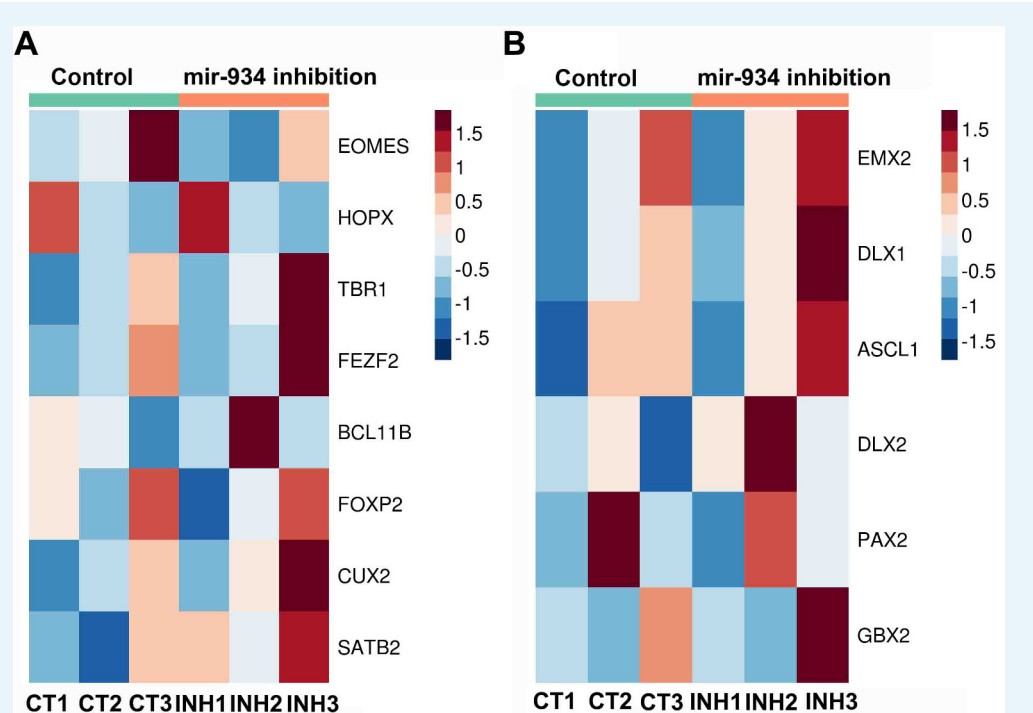

**Appendix 1—figure 4.** Heatmaps showing the expression of genes displaying no significant differences following miR-934 sustained inhibition. (**A**) Genes associated with intermediate progenitors, outer radial glia and cortical plate neurons. (**B**) Genes associated with regional identities during brain development.

## Dual luciferase assays for identification of the binding site of miR934 on the 3'-UTR of Fzd5, TFCP2L1 and RAB3B

The full sequences of the two predicted miRNA recognition elements (MREs) of miR-934 on the 3'-UTR of Fzd5 are the following, with the seed-matched sequences underlined: MRE (7-mer): TAGAGCCAGCGATACTTTATTT**GTAGACA**; MRE (8-mer): TTTCATATGTCTTCCCA-GAAAA**GTAGACA**. DNA fragments of the 3'-UTR of the human Frizzled class Receptor 5 (FZD5) were cloned by PCR, using human genomic DNA isolated from human fibroblasts. Primers were designed based on NM_003468.3 NCBI Reference Sequence with the knowledge that the 3'-UTR mRNA sequence of the human FZD5 derives from a single exon. The 1012 bp DNA fragment containing the 7mer sequence was produced using primers F2 and R2, and was subsequently cloned into SacI and XbaI restriction sites of the pmirGLO vector. The F3 and R1 primers were used for cloning the 808 bp DNA fragment containing the 8mer sequence. The PCR product was cloned into NheI and XbaI restrictions sites of the pmirGLO vector.

The full sequences of the two predicted miRNA recognition elements (MREs) of miR-934 on the 3'-UTR of the human transcription factor CP2-like 1 (TFCP2L1) are the following, with the seed-matched sequences underlined: MRE (7-mer): AGCCCTGTGGATCCCCGTGGAT**G TAGACA**; MRE (9-mer): AAAAAAAAAAATTCTTATTTT**TAGTAGACA**. DNA fragments corresponding to the 3'-UTR of the human TFCP2L1 mRNA, were cloned by PCR, using human genomic DNA isolated from human fibroblasts. Both of the above MRE sequences are located at exon 15 of human TFCP2L1 gene. Primers were designed based on NM_014553.3 NCBI Reference Sequence. PCR products of 218 bp and 332 bp containing the 7-mer and 9-mer sequences respectively, were produced using primers F4, R4 for the 7-mer and F5, R5 for the 9-mer and were subsequently cloned separately into SacI and XbaI restriction sites of the pmirGLO vector. The seed-matched sequence corresponding to MRE (9mer) of TFCP2L1 was mutated to TA**TTCTGAT** using the Q5 Site-Directed Mutagenesis Kit (NEB) and the primers

FOR mut and REV mut while DNA sequencing was performed for validation of mutated seed sites.

The full sequences of the two predicted miRNA recognition elements (MREs) of miR-934 on the 3'-UTR of the human RAB3B are the following, with the seed-matched sequences underlined: MRE (8-mer): AGGTCCTGATCCATTATATGA**AGTAGACA**; MRE (7-mer): AG TAAACTATCTTCCAACCCCT**GTAGACA**. DNA fragments corresponding to the 3'-UTR of the human RAB3B mRNA, were cloned by PCR, using human genomic DNA isolated from human fibroblasts. Both of the above MRE sequences are located at exon 5 of human RAB3B gene and primers were designed based on NM_002867.4 NCBI Reference Sequence. PCR products of 339 bp and 342 bp containing the 8-mer and 7-mer sequences respectively, were produced using primers F6, R6 for the 8-mer and F7, R7 for the 7-mer and were subsequently cloned separately into SacI and XbaI restriction sites of the pmirGLO vector.

All clones were verified by sequencing analysis. All primers used for cloning and mutagenesis are listed in *Supplementary file 7*. HEK293T cells were co-transfected with the modified constructs and miRNA mimics (100 nM, Exiqon/Qiagen) and luciferase activity was measured 48 hr later with the Promega Luciferase assay system.

