## [Decision Letter]

**Acceptance summary:**

Prodromidou and colleagues have performed a nice study utilizing induced neural progenitor cell (NPC) to demonstrate an important role for a primate-specific mir-934 in early human neurogenesis The work is very interesting and well-presented, highlighting the evolutionary role on this microRNA which plays a role in human and non-human primates neurogenesis.

**Decision letter after peer review:**

Thank you for submitting your article "MicroRNA-934 is a novel primate-specific small non-coding RNA with neurogenic function" for consideration by *eLife*. Your article has been reviewed by three peer reviewers, and the evaluation has been overseen by Marianne Bronner as the Senior and Reviewing Editor. The following individual involved in review of your submission has agreed to reveal their identity: Idoia Quintana-Urzainqui (Reviewer #3).

The reviewers have discussed the reviews with one another and the Reviewing Editor has drafted this decision to help you prepare a revised submission.

Summary:

In this work, Matsas and colleagues identify and characterize miR-934 as a primate-specific and progenitor stage specific microRNA during human pluripotent cell differentiation. Using RNA-seq, the authors detect that miR-934 is transiently expressed at the neuronal progenitor amplification state. They go on to test its role by a series of functional experiments and show that it promotes differentiation over proliferation and identify some of its potential targets.

Essential revisions:

The reviewers agree that the manuscript is potentially exciting and of interest for eventual publication in *eLife*. However, they feel that the manuscript requires further data analysis and to fully support the conclusions. For example, you need to:

1) present a full list of miRNAs expressed at different stages and discuss how/why you have selected the mir934 and their targets from the bulk data.

2) show the full list of targets identified after miR934 inhibition and the criteria used to select the four that are presented.

3) provide more data to support the enrichment of subplate genes compared to other regions or compartments of the developing cortex (iSVZ, oSVZ, inner layers).

The full reviews are attached below for further details.

Reviewer #1:

Prodromidou and colleagues presented a nice piece of work demonstrating the role of a primate-specific mir-934 which is necessary for the early human neurogenesis utilizing induced neural progenitor cell (NPC). The work is very interesting and well-presented, highlighting the evolutionary role on this microRNA having an exclusive role on human and non-human primates 'neurogenesis. However, for this reviewer there are some concerns that need to be addressed before publication.

1) The author should present the complete list of microRNAs that are expressed at the different stages. Is miR934 the only one highly expressed at the NPCs stage?

2) The authors used RNA-seq to identify miR-934 target, however many microRNA doesn't affect the expression but affect the translation. This needs to be mentioned in the Results/Discussion.

3) It would be desired that the authors provide a complete list of miR-934 targets and their transcriptional variation after miR934 inhibition. It is not very clear if the four presented genes (STMN2, TFCP2L1, RAB3B and FZD5) are the only one presenting variation or if there are other potential targets having variations. In the same context, it is difficult to explain that having only four targets the authors observed a very broad transcriptional changes (1458 genes and 79 microRNAs). Base on this, have STMN2, TFCP2L1, RAB3B and FZD5 a very high hierarchy in the gene regulatory network that control human neurogenesis to explain the observed changes? If this is the case, it would be nice to do rescue experiment by restoring the functionality of those genes by using overexpressing constructs.

4) The effect of miR934 inhibition on the increasing of FZD5 protein is not very obvious from the presented WB. Moreover, it is intriguing for this reviewer that the authors presented qPCR for the other three genes and WB for FZD5. On the other side it would be nice to demonstrate the effect of overexpressing the miR934 on the expression of the targets genes.

5) The gold standard to validate the direct interaction of specific microRNAs with putative target genes is to use the entire 3'UTRs. In this case the authors only used 29nt which wouldn't represent the real 3D context of the entire 3'UTR. Moreover, they only test for one gene (they should test for the other 3 presented as a putative direct targets) and the nucleotide changes generated on the mutated form is to extreme. Generally, 2 to 3 bases on the seeds are sufficient to affect the specific binding of the microRNA without affecting the stability of the mRNA (see Ryan et al., 2016 BMC Bioinf to generate the specific mutation on the seed sequence).

Reviewer #2:

In this manuscript, Prodromidou et al. propose that the primate specific microRNA miR-934 is a novel developmental regulator of neurogenesis. Using neural induction of stem cell cultures as a model for human brain development, they argue that miR-934 acts at the progenitor-to-neuroblast transition to regulate genes associated with progenitor cell proliferation and differentiation.

Previous work has identified many primate specific miRNAs with functions in brain development, as well as shown that some of these miRNAs have expression patterns specific to particular brain regions. Along these lines, the authors suggest that miRNA-934 specifically affects genes characteristic of the subplate, a transient brain region unique to primates. This claim is not investigated further and its relevance seems tenuous.

Overall, this manuscript provides compelling evidence that miRNA-934 is involved in regulation of genes associated with cell proliferation and neuronal differentiation during progenitor-to-neuroblast transition in primates. It could also benefit from some revisions.

1) Figure 1A: It would be nice to see panels of all of these markers at all stages of differentiation to show how much heterogeneity/overlap of stages there is during the differentiation process.

2) Figure 4E: The western blot shown here is not of sufficient quality to convincingly support the conclusion that FZD5 and active β-catenin are upregulated upon miR-934 inhibition. It should be cleaner (FZD5) and less saturated (β-catenin) to be able to properly quantify relative protein levels. Based on Figure 4C, <50% of cells are experiencing miR-934 inhibition in a given sample. It would be helpful to show by immunofluorescence that GFP positive cells show an increase in FZD5 and translocation of β-catenin to the nucleus compared to non-transfected cells.

3) Figure 5F: The authors "compared differential RNA expression data with genes that characterize the molecular profile of prodromal/pioneer neuronal populations" in order to "determine the developmental stage involved," however they only analyze two populations-"deep layer neurons" for which the data is not shown, and the subplate. The data shown in this figure gives the absolute value of fold change for these genes when miR-934 is inhibited, which is strange when all the other heatmaps are based on directional fold changes. Another explanation for the change seen in "subplate enriched genes" is that many genes involved in many regions of the developing brain are altered downstream of perturbation during progenitor-to-neuroblast transition. To substantiate their claim the authors would need to show that genes characteristically involved in the development of several other regions of the developing brain are not dysregulated upon miR-934 inhibition. Ideally, they would also show that endogenous miR-934 is differentially expressed in the subplate compared to other brain regions. Without this validation, this data appears cherry picked and does not add anything of value to the paper.

Reviewer #3:

In general, this is a well-executed work, with potential impact on the biology and evolution of the neocortex. My main concerns have to do with the lack of clarity in the way the results are sometimes presented, which I will detail in the following sections.

Regarding the first section of the Results, I think the characterization of the stages of differentiation of their cultured systems is convincing.

The characterization of miR-934 as a species- and stage-specific miRNA is a key point in this paper, therefore, I would like to see more detail on the analysis that led to these conclusions. The lack of clarity sometimes impedes me from assessing the real impact of the findings. Main concerns:

– Subsection “Identification of miR-934 with species- and stage-specific expression during progenitor expansion and early neuron generation”, second paragraph. The authors indicate how they identified miR934 as a high and segregated expressed miRNA, associated to the stage of neural induction. This expression is very specific and convincing, but could they give more details on how did they select this gene? Was is the only one showing this stage-specificity? How many others were included in the analysis? This will certainly help to convince the reader on how specific and, therefore, potentially important this micro RNA is for neural induction.

– To determine the species-specificity the authors perform an in silico analysis and detect that miR-934 is only annotated in certain primates. Have the authors analysed its presence/absence in other species or animal groups? It would add value to the discovery, to include analysis in other non-primate species and even non-mammalian, showing its absence.

– The authors show that sustained inhibition alters the expression of subplate-enriched genes and they report no difference in deep layer associated genes. Have they analysed other compartments of the developing cortex? I think they should indicate this. In particular, it would be very interesting to see whether there is any enrichment in subventricular zone/outer subventricular zone associated genes. Since they analysed deep layer genes, they could also mention if they find any change in upper layers associated genes.

[Editors' note: further revisions were suggested prior to acceptance, as described below.]

Thank you for resubmitting your work entitled "MicroRNA-934 is a novel primate-specific small non-coding RNA with neurogenic function during early development" for further consideration by *eLife*. Your revised article has been reviewed by two peer reviewers and the evaluation has been overseen by Marianne Bronner as the Senior and Reviewing Editor.

While the manuscript has been improved, there are some remaining issues that need to be addressed before acceptance. I refer you to the reviewers' comments for clarification.

Reviewer #1:

Prodromidou and colleagues presented a new version of the manuscript demonstrating the role of a primate-specific mir-934 which is necessary for the early human neurogenesis utilizing induced neural progenitor cell (NPC). Although the authors attempt to answer most to this reviewer concerns, there are still several important points that were not addressed and are essential to support the major conclusions of the work.

In the requested list of miR-934 putative targets affected after miR934 inhibition FZD5 is not present and there are only 3 genes (STMN2, TFCP2L1, and RAB3B) (Supplementary file 5). The lack of changes in FZD5 is in agreement with the lack of a very obvious change on the WB assay (the picture is still not very convincing) and the modest change observed by qPCR (less than 1.5 fold) after miR934 inhibition. It is still not clear how the changes on these three genes may have a very broad transcriptional changes. However, based on what the author answered saying that these 3 genes have a very high hierarchy it would be very interesting that the authors make rescue experiment by adding those factor back after miR934 inhibition, but the authors never included in their new version this experiment or have discussed why this is not feasible.

Finally, the authors claim in the Discussion that they have identified four miR-934 targets (TFCP2L1, FZD5, STMN2 and RAB3B), however they partially demonstrated the direct association with Fzd5, the less affected after mir934 inhibition on the qPCR analyzes, even when this reviewer specifically requested the demonstration of this direct association for the other three factors by using luciferase assay. This experiment should be performed by using the WT and mutated 3'UTRs (or regions of the 3'UTRs in case those are extremely big) of the target genes.

Reviewer #2:

In this revised manuscript, Prodromidou et al. addressed essential revisions and provided explanations for reviewer suggested revisions they did not carry out. Overall, most issues were addressed satisfactorily and the added material bolsters their conclusions, as well as increases the clarity of the paper. Issues remain, however, with Figure 4. Though the authors have added FZD5 qPCR data to support their conclusion that FZD5 is upregulated on miR-934 depletion, the western blot presented here remains problematic. The quality is insufficient to convincingly support their claim that FZD5 protein levels and activated B-catenin are increased, and this is the only data they present to support their conclusion that miR-934 inhibition of FZD5 leads to a reduction of canonical Wnt signaling. The perceived quality of the paper would benefit from a cleaner/less saturated western blot, or perhaps a reporter based assay to show that Wnt signaling is affected. I recommend this paper for acceptance once this revision is carried out.

---

## [Author Response]

Essential revisions:The reviewers agree that the manuscript is potentially exciting and of interest for eventual publication in eLife. However, they feel that the manuscript requires further data analysis and to fully support the conclusions. For example, you need to:1) present a full list of miRNAs expressed at different stages and discuss how/why you have selected the mir934 and their targets from the bulk data.

We present new data showing how we have selected miRNA-934. In particular, we explain (Results subsection “Identification of miR-934 with species- and stage-specific expression during progenitor expansion and early neuron generation”) that for this purpose, we adopted the tissue-specificity index tau, which is a reliable method for estimation of gene expression specificity and has been shown to outperform other relevant indices (Kryuchkova-Mostacci and Robinson-Rechavi, 2017). Calculation of tau index across the distinct differentiation phases revealed that 144 miRNAs exhibited a highly specific expression profile (tau>0.7) (revised Figure 2B and Supplementary file 1A included in Supplementary file 1). For example using this computational method, miR-302 that is a known marker of pluripotency, was allocated at the hESC/iPSC stage.

Further we show that when these 144 miRNAs exhibiting tissue specificity (tau > 0.7) were ranked by decreasing order of median expression in NPCs, miR-934 emerged on top of the list (revised Figure 2B), scoring the highest expression among other miRNAs at the NPC stage and demonstrating a strikingly segregated expression as compared to all other stages (tau value 0.76) (new Supplementary file 1). Moreover, according to median expression differences, no other miRNA was identified to have such a clear disparity against other stages.

Regarding species conservation for miR-934, we performed two distinct in silico analyses to identify species having potentially mature miRNAs with similar sequences. Both queries identified only primate species having a mature miRNA or miRNA precursor with similarity (e-value <1) to the mature hsa-miR-934 sequence. The methodology used is now included in Appendix 1 and the results are described in the Results section (subsection “Identification of miR-934 with species- and stage-specific expression during progenitor expansion and early neuron generation”) and presented in Supplementary file 2.

2) show the full list of targets identified after miR934 inhibition and the criteria used to select the four that are presented.

Regarding miRNA-934 target identification, as we already explained in our original manuscript, their selection was made by exploring the RNA-seq data obtained upon transition from hESCs/iPSCs to NPCs, as well as a second set of RNA-seq data generated at the stage of neural induction of hESCs following sustained inhibition of miR-934 function via transduction with the miRZip lentivector-based anti-microRNA system. To identify miR-934 targets in each case, we integrated small RNA and RNA sequencing data using the algorithm presented in mirExTra v2. Targets were detected using microT-CDS target prediction tool as the source of potential interactions.

Using this approach we identified the four miR-934 targets presented in our original manuscript, comprising Frizzled 5 (FZD5), TFCP2L1, STMN2 (stathmin-2, SCG10) and RAB3B. Then we confirmed by qRT-PCR that sustained inhibition of miR-934 during neural induction up-regulates the 4 predicted targets FZD5, TFCP2L1, STMN2, and RAB3B. The *in silico* analysis also predicted F11R a junctional adhesion molecule and SLC16A1 a monocarboxylate transporter as potential binding partners of miR-934, yet we could not confirm changes in their mRNA expression upon miRNA-934 perturbation. This information is now included in the Results subsection “The mRNA targets of miR-934 during neural induction are associated with progenitor proliferation and neuronal differentiation” and in the revised Figure 4. No other targets, apart from the aforementioned, were predicted for miRNA-934.

3) provide more data to support the enrichment of subplate genes compared to other regions or compartments of the developing cortex (iSVZ, oSVZ, inner layers).

We now present additional data highlighting the role of miR-934 in modulating molecular pathways mediating neurogenic events associated with early dorsal progenitor populations and the subplate zone (Results subsection “Inhibition of miR-934 during neural induction affects molecular pathways of neurogenesis and alters the expression of subplate-enriched genes”).

In this regard, we attest that apart from the genes reported in our original manuscript regarding cellular identities and the subplate region (Figure 5F and validation by qRT-PCR in Figure 5G; Supplementary file 7), we did not observe any differences in the expression of the transcription factor Tbr2 (EOMES) that characterizes intermediate progenitors or the outer radial glia marker HOPX (now shown in new Appendix 1—figure 3A). Moreover, we indicate that there were no changes in ventral or mid-brain fate specification markers, including DLX1, DLX2, ALSC1, PAX2 and GBX2 which all exhibited lower expression as compared to the dorsal forebrain marker EMX2 (new Appendix 1—figure 3B). Of interest, we did not detect expression of early markers of lateral and medial ganglionic eminences, such as Nkx2.1, GSX1 and GSX2 with or without miR-934 perturbation.

Further to determine the developmental stage involved, we compared differential RNA expression data for genes that characterize the molecular profile of prodromal/pioneer neuronal populations during cortical development. We detected low expression in a series of early neuronal markers, while further analysis did not reveal miR-934-mediated differences in the expression of genes characterizing deep layer neurons, such as FEZF2, TBR1, FOXP2 and CTIP2 (BCL11B), or the early fate determinant of upper layer neurons CUX2 (51) (Appendix 1—figure 3A). We detected expression of SATB2, which is reported to be co-expressed with CTIP2 in a subset of early neurons of the visual cortex, yet this transcription factor presented no change upon miR-934 perturbation (Appendix 1—figure 3A).

This data further support the role of miR-934 in modulating molecular pathways mediating neurogenic events associated with early dorsal progenitor populations and the subplate zone.

The full reviews are attached below for further details.Reviewer #1:Prodromidou and colleagues presented a nice piece of work demonstrating the role of a primate-specific mir-934 which is necessary for the early human neurogenesis utilizing induced neural progenitor cell (NPC). The work is very interesting and well-presented, highlighting the evolutionary role on this microRNA having an exclusive role on human and non-human primates 'neurogenesis. However, for this reviewer there are some concerns that need to be addressed before publication.1) The author should present the complete list of microRNAs that are expressed at the different stages. Is miR934 the only one highly expressed at the NPCs stage?

We have addressed this comment, please see essential revisions above.

2) The authors used RNA-seq to identify miR-934 target, however many microRNA doesn't affect the expression but affect the translation. This needs to be mentioned in the Results/Discussion.

We have added a relevant comment in the second paragraph of the Discussion.

3) It would be desired that the authors provide a complete list of miR-934 targets and their transcriptional variation after miR934 inhibition. It is not very clear if the four presented genes (STMN2, TFCP2L1, RAB3B and FZD5) are the only one presenting variation or if there are other potential targets having variations. In the same context, it is difficult to explain that having only four targets the authors observed a very broad transcriptional changes (1458 genes and 79 microRNAs). Base on this, have STMN2, TFCP2L1, RAB3B and FZD5 a very high hierarchy in the gene regulatory network that control human neurogenesis to explain the observed changes? If this is the case, it would be nice to do rescue experiment by restoring the functionality of those genes by using overexpressing constructs.

We have addressed the comment regarding the full list of target identification, please see essential revisions above.

Regarding the reviewer’s comment on the brad transcriptional changes observed upon miRNA-934 sustained inhibition, we have added a comment in the second paragraph of the Discussion arguing that the identified targets play a role in diverse processes along the neurogenic program including stem cell self-renewal, neural progenitor proliferation and differentiation as well as cytoskeletal arrangement and neurotransmission. It is therefore conceivable that miR-934-mediated modulation in the expression of these molecules should further influence downstream pathways related to morphological and functional transition of neural progenitor cells, thus causing an amplification in gene expression changes. Consistently, we observed altered expression in a total of 1458 genes upon miR-934 inhibition.

We also argue that the downstream implications of miR-934 on post-transcriptional control of gene expression may be even broader since miRNAs, besides causing mRNA degradation, may also act to inhibit translation of their mRNA targets thus affecting expression at the protein level.

4) The effect of miR934 inhibition on the increasing of FZD5 protein is not very obvious from the presented WB. Moreover, it is intriguing for this reviewer that the authors presented qPCR for the other three genes and WB for FZD5. On the other side it would be nice to demonstrate the effect of overexpressing the miR934 on the expression of the targets genes.

We now present qRT-PCR results for FZD5.

We feel that sustained inhibition of miRNA-934 is more sensitive for assessing the effects on target genes, since often overexpression may not have the expected influence, particularly if levels are already saturated.

5) The gold standard to validate the direct interaction of specific microRNAs with putative target genes is to use the entire 3'UTRs. In this case the authors only used 29nt which wouldn't represent the real 3D context of the entire 3'UTR. Moreover, they only test for one gene (they should test for the other 3 presented as a putative direct targets) and the nucleotide changes generated on the mutated form is to extreme. Generally, 2 to 3 bases on the seeds are sufficient to affect the specific binding of the microRNA without affecting the stability of the mRNA (see Ryan et al., 2016 BMC Bioinf to generate the specific mutation on the seed sequence).

We have tested binding particularly for Fzd5 in order to assess whether binding of miR-934 involves a conserved site on the 3’UTR of Fzd5. In this case, we performed experiments by cloning the respective 29 nt and also by cloning fragment of the entire 3’UTR containing the respective site. Since the 3’UTR of Fzd5, which is a particularly large 3’UTR, contains 2 candidate sites for interaction with miR-934, we used separate fragments of the 3’UTR for cloning in order to distinguish the 2 binding sites and identify the true region of interaction with miR-934.

Reviewer #2:[…] Overall, this manuscript provides compelling evidence that miRNA-934 is involved in regulation of genes associated with cell proliferation and neuronal differentiation during progenitor-to-neuroblast transition in primates. It could also benefit from some revisions.1) Figure 1A: It would be nice to see panels of all of these markers at all stages of differentiation to show how much heterogeneity/overlap of stages there is during the differentiation process.

We appreciate the comment of this reviewer. However, we feel that our global transcriptomic data show clearly that cells sequester primarily according to differentiation stage, a fact that we also observe by immunocytochemistry for cell-type-specific markers.

2) Figure 4E: The western blot shown here is not of sufficient quality to convincingly support the conclusion that FZD5 and active β-catenin are upregulated upon miR-934 inhibition. It should be cleaner (FZD5) and less saturated (β-catenin) to be able to properly quantify relative protein levels. Based on Figure 4C, <50% of cells are experiencing miR-934 inhibition in a given sample. It would be helpful to show by immunofluorescence that GFP positive cells show an increase in FZD5 and translocation of β-catenin to the nucleus compared to non-transfected cells.

We now also present qRT-PCR results for FZD5 confirming its up-regulation upon miR-934 inhibition.

Unfortunately we could not perform the experiment suggested by this reviewer for β-catenin translocation because our available antibody does not give a clear picture by immunofluorescence.

3) Figure 5F: The authors "compared differential RNA expression data with genes that characterize the molecular profile of prodromal/pioneer neuronal populations" in order to "determine the developmental stage involved," however they only analyze two populations-"deep layer neurons" for which the data is not shown, and the subplate. The data shown in this figure gives the absolute value of fold change for these genes when miR-934 is inhibited, which is strange when all the other heatmaps are based on directional fold changes. Another explanation for the change seen in "subplate enriched genes" is that many genes involved in many regions of the developing brain are altered downstream of perturbation during progenitor-to-neuroblast transition. To substantiate their claim the authors would need to show that genes characteristically involved in the development of several other regions of the developing brain are not dysregulated upon miR-934 inhibition. Ideally, they would also show that endogenous miR-934 is differentially expressed in the subplate compared to other brain regions. Without this validation, this data appears cherry picked and does not add anything of value to the paper.

We have addressed this comment, please see essential revisions above.

Reviewer #3:In general, this is a well-executed work, with potential impact on the biology and evolution of the neocortex. My main concerns have to do with the lack of clarity in the way the results are sometimes presented, which I will detail in the following sections.Regarding the first section of the Results, I think the characterization of the stages of differentiation of their cultured systems is convincing.The characterization of miR-934 as a species- and stage-specific miRNA is a key point in this paper, therefore, I would like to see more detail on the analysis that led to these conclusions. The lack of clarity sometimes impedes me from assessing the real impact of the findings. Main concerns:– Subsection “Identification of miR-934 with species- and stage-specific expression during progenitor expansion and early neuron generation”, second paragraph. The authors indicate how they identified miR934 as a high and segregated expressed miRNA, associated to the stage of neural induction. This expression is very specific and convincing, but could they give more details on how did they select this gene? Was is the only one showing this stage-specificity? How many others were included in the analysis? This will certainly help to convince the reader on how specific and, therefore, potentially important this micro RNA is for neural induction.

We have addressed this comment, please see essential revisions above.

– To determine the species-specificity the authors perform an in silico analysis and detect that miR-934 is only annotated in certain primates. Have the authors analysed its presence/absence in other species or animal groups? It would add value to the discovery, to include analysis in other non-primate species and even non-mammalian, showing its absence.

We have addressed this comment, please see essential revisions above.

– The authors show that sustained inhibition alters the expression of subplate-enriched genes and they report no difference in deep layer associated genes. Have they analysed other compartments of the developing cortex? I think they should indicate this. In particular, it would be very interesting to see whether there is any enrichment in subventricular zone/outer subventricular zone associated genes. Since they analysed deep layer genes, they could also mention if they find any change in upper layers associated genes.

We have addressed this comment, please see essential revisions above.

[Editors' note: further revisions were suggested prior to acceptance, as described below.]

While the manuscript has been improved, there are some remaining issues that need to be addressed before acceptance. I refer you to the reviewers' comments for clarification.Reviewer #1:Prodromidou and colleagues presented a new version of the manuscript demonstrating the role of a primate-specific mir-934 which is necessary for the early human neurogenesis utilizing induced neural progenitor cell (NPC). Although the authors attempt to answer most to this reviewer concerns, there are still several important points that were not addressed and are essential to support the major conclusions of the work.In the requested list of miR-934 putative targets affected after miR934 inhibition FZD5 is not present and there are only 3 genes (STMN2, TFCP2L1, and RAB3B) (Supplementary file 5). The lack of changes in FZD5 is in agreement with the lack of a very obvious change on the WB assay (the picture is still not very convincing) and the modest change observed by qPCR (less than 1.5 fold) after miR934 inhibition. It is still not clear how the changes on these three genes may have a very broad transcriptional changes. However, based on what the author answered saying that these 3 genes have a very high hierarchy it would be very interesting that the authors make rescue experiment by adding those factor back after miR934 inhibition, but the authors never included in their new version this experiment or have discussed why this is not feasible.

We have now improved the quality of the immunoblot in Figure 4 relating to FZD5 protein levels, clearly showing that Fzd5 protein is upregulated upon miR934 inhibition.

Regarding the rescue experiments and following correspondence with the Editor, it has been agreed that these are very challenging both in technical terms and also in faithfully simulating the pleiotropic spatiotemporal control of miRNAs, exceeding the scope of the present manuscript.

Finally, the authors claim in the Discussion that they have identified four miR-934 targets (TFCP2L1, FZD5, STMN2 and RAB3B), however they partially demonstrated the direct association with Fzd5, the less affected after mir934 inhibition on the qPCR analyzes, even when this reviewer specifically requested the demonstration of this direct association for the other three factors by using luciferase assay. This experiment should be performed by using the WT and mutated 3'UTRs (or regions of the 3'UTRs in case those are extremely big) of the target genes.

We have generated 4 modified constructs of the dual luciferase reporter pmirGLO vector following cloning of 2 fragments of the 3'-UTR of TFCP2L1 containing each of the two possible binding domains for miR-934 (218 bp, including the 7mer sequence; and 339 bp including the 9-mer sequence) and cloning of another 2 fragments of the 3'-UTR of RAB3B containing each of the two possible binding domains for miR-934 (339 bp, including the 8mer sequence; and 342 bp including the 7-mer sequence). A mutated version of the seed-matched sequence corresponding to 9mer containing MRE of TFCP2L1 was also generated and the respective construct was cloned into the pmiRGLO vector serving as a control for the luciferase assay (Appendix 1 for methodology). Co-transfection of HEK293T cells with miR-934 mimics and the different TFCP2L1 and RAB3B reporter constructs suppressed luciferase activity in both constructs for TFCP2L1 (p=0.0002 for 7mer and p=0.0007 for 9mer) and only in the construct containing the 8mer site sequence for RAB3B (p=0.002). By contrast luciferase activity was not affected upon transfection of control constructs (pmiRGLO vector and the mutated TFCP2L1 9mer) in the presence of miR934 mimics. These data confirm TFCP2L1 and RAB3B as binding partners of miR934 and validate specific predicted sites as regions of interaction. (subsection “The mRNA targets of miR-934 during neural induction are associated with progenitor proliferation and neuronal differentiation”, new Appendix 1—figure 2 and Appendix 1 for figure legend).

Unfortunately due to the lockdown, we were unable to produce additional constructs required for validation of STMN2 as a direct miR934 target and the remaining mutated constructs, but we hope that the reviewer will find this acceptable given the circumstances, also considering that all performed validation assays confirmed bioinformatics data for target prediction, as well as taking into account the complete set of data presented in our article.

Reviewer #2:In this revised manuscript, Prodromidou et al. addressed essential revisions and provided explanations for reviewer suggested revisions they did not carry out. Overall, most issues were addressed satisfactorily and the added material bolsters their conclusions, as well as increases the clarity of the paper. Issues remain, however, with Figure 4. Though the authors have added FZD5 qPCR data to support their conclusion that FZD5 is upregulated on miR-934 depletion, the western blot presented here remains problematic. The quality is insufficient to convincingly support their claim that FZD5 protein levels and activated B-catenin are increased, and this is the only data they present to support their conclusion that miR-934 inhibition of FZD5 leads to a reduction of canonical Wnt signaling. The perceived quality of the paper would benefit from a cleaner/less saturated western blot, or perhaps a reporter based assay to show that Wnt signaling is affected. I recommend this paper for acceptance once this revision is carried out.

As stated above we have now improved the quality of the immunoblot in Figure 4 relating to FZD5 and β-catenin protein levels, clearly showing that Wnt signaling is affected.